# High-performance ferroelectric field-effect transistors with ultra-thin indium tin oxide channels for flexible and transparent electronics

Qingxuan Li [1,2,4] ✉, Siwei Wang[2,4], Zhenhai Li[2], Xuemeng Hu[2], Yongkai Liu[2], Jiajie Yu[2], Yafen Yang [2], Tianyu Wang [2], Jialin Meng[2], Qingqing Sun[2], David Wei Zhang[2] & Lin Chen [2,3] ✉

With the development of wearable devices and hafnium-based ferroelectrics (FE), there is an increasing demand for high-performance flexible ferroelectric memories. However, developing ferroelectric memories that simultaneously exhibit good flexibility and significant performance has proven challenging. Here, we developed a high-performance flexible field-effect transistor (FeFET) device with a thermal budget of less than 400 °C by integrating Zr-doped $HfO_2$ (HZO) and ultra-thin indium tin oxide (ITO). The proposed FeFET has a large memory window (MW) of 2.78 V, a high current on/off ratio ($I_{ON}/I_{OFF}$) of over $10^8$, and high endurance up to $2\times10^7$ cycles. In addition, the FeFETs under different bending conditions exhibit excellent neuromorphic properties. The device exhibits excellent bending reliability over $5\times10^5$ pulse cycles at a bending radius of 5 mm. The efficient integration of hafnium-based ferroelectric materials with promising ultrathin channel materials (ITO) offers unique opportunities to enable high-performance back-end-of-line (BEOL) compatible wearable FeFETs for edge intelligence applications.

The discovery of hafnium-based ferroelectrics (FE) promoted the development of ferroelectric field-effect transistors (FeFETs), which have ultra-low power consumption, fast erasing speed and strong scalability, making them occupy a place in the field of non-volatile memory (NVM)[1–5]. The channel conductance of the transistor is controlled by polarization switching of the gate ferroelectric barrier, thereby achieving fast read/write operations of the device[6,7]. Since the preparation process of FeFETs requires high temperature, it is difficult to be compatible with flexible substrates, which limits its application in the wearable field. Until now, FeFETs based on rigid substrates have exhibited superior performance. Although ferroelectric devices fabricated on flexible substrates, such as systems based on polyvinylidene

fluoride (PVDF) and its copolymer polyvinylidene fluoride-trifluoroethylene (PVDF-TrFE), have also demonstrated excellent capabilities, proven to exceed $10^8$ cycles of endurance[8–10], they are confronted with limitations of film thickness and driving voltage[11–14]. Moreover, the requirements for the use scenarios are strict, and the performance in the bending state is difficult to be guaranteed, which limits their practical applications.

And with the scaling of transistor sizes, there is a higher demand for ultra-thin channel materials. For example, two-dimensional van der Waals materials have received more attention[15,16]. However, these materials usually have high contact resistance with metal electrodes and are easy to form interface traps. More importantly, their

[1]School of Integrated Circuits, Anhui University, Anhui 230601, PR China. [2]School of Microelectronics, Fudan University, Shanghai 200433, PR China. [3]Zhangjiang Fudan International Innovation Center, Shanghai 201203, PR China. [4]These authors contributed equally: Qingxuan Li, Siwei Wang. ✉ e-mail: liqx@ahu.edu.cn; linchen@fudan.edu.cn

preparation processes are not compatible with CMOS[17]. Studies have shown that oxide semiconductors commonly used as transparent contact layers such as amorphous indium gallium zinc oxide (IGZO) and indium tin oxide (ITO) have become promising candidates for ultra-thin channels[18–20]. They can achieve monolithic 3-D integration, have high carrier mobility, high uniformity wafer-level deposition film formation, excellent flexibility, and their low process temperature overcomes the back-end-of-line (BEOL) low thermal budget process limitations[21,22]. According to previous reports, the carrier density controllability of the ITO channel can be effectively enhanced through thickness scaling and ferroelectric gating[19,22,23]. P(VDF-TrFE) is cost-effective, easy to fabricate via sol-gel processing, and requires a low processing temperature, making it a promising ferroelectric material for FeFETs and synaptic devices[11,24–26]. However, P(VDF-TrFE) films are typically thick (over 100 nm) and necessitate higher driving voltages for polarization switching and saturation[11–14]. In contrast, thin Zr-doped HfO₂ (HZO) exhibits excellent ferroelectric performance at low driving voltages even at a thickness of merely 5nm[27–29], which is crucial for

scalable integrated circuits incorporating ultra-thin oxide semi-conductors. Currently, whether it is two-terminal devices like FRAM or three-terminal transistors integrated with IGZO channels, the anneal-ing temperature of HZO has been shown to be compatible with BEOL conditions[19,30–32]. In addition, FeFETs based on HZO have demon-strated synaptic characteristics[33]. On the other hand, the ferroelectric properties of HZO films fabricated on flexible mica substrates have continued to exhibit exceptional performance[34]. Nonetheless, achiev-ing high-performance HZO-based ferroelectric memories on flexible substrates remains challenging. Since ferroelectric materials such as Zr-doped HfO₂ (HZO) have good switchable polarization, when used as gate ferroelectric dielectrics, they can effectively modulate carriers in the ITO channel, resulting in excellent performance[3,35,36].

In this work, we report a BEOL-compatible high-performance FeFET memory device based on HZO and ultrathin ITO channels on a flexible MICA substrate, which outperforms rigid substrates. The proposed FeFET has a minimum subthreshold swing (S.S.) of 33 mV/decade, a current on/off ratio ($I_{ON}/I_{OFF}$) ratio of over $10^8$, and a very

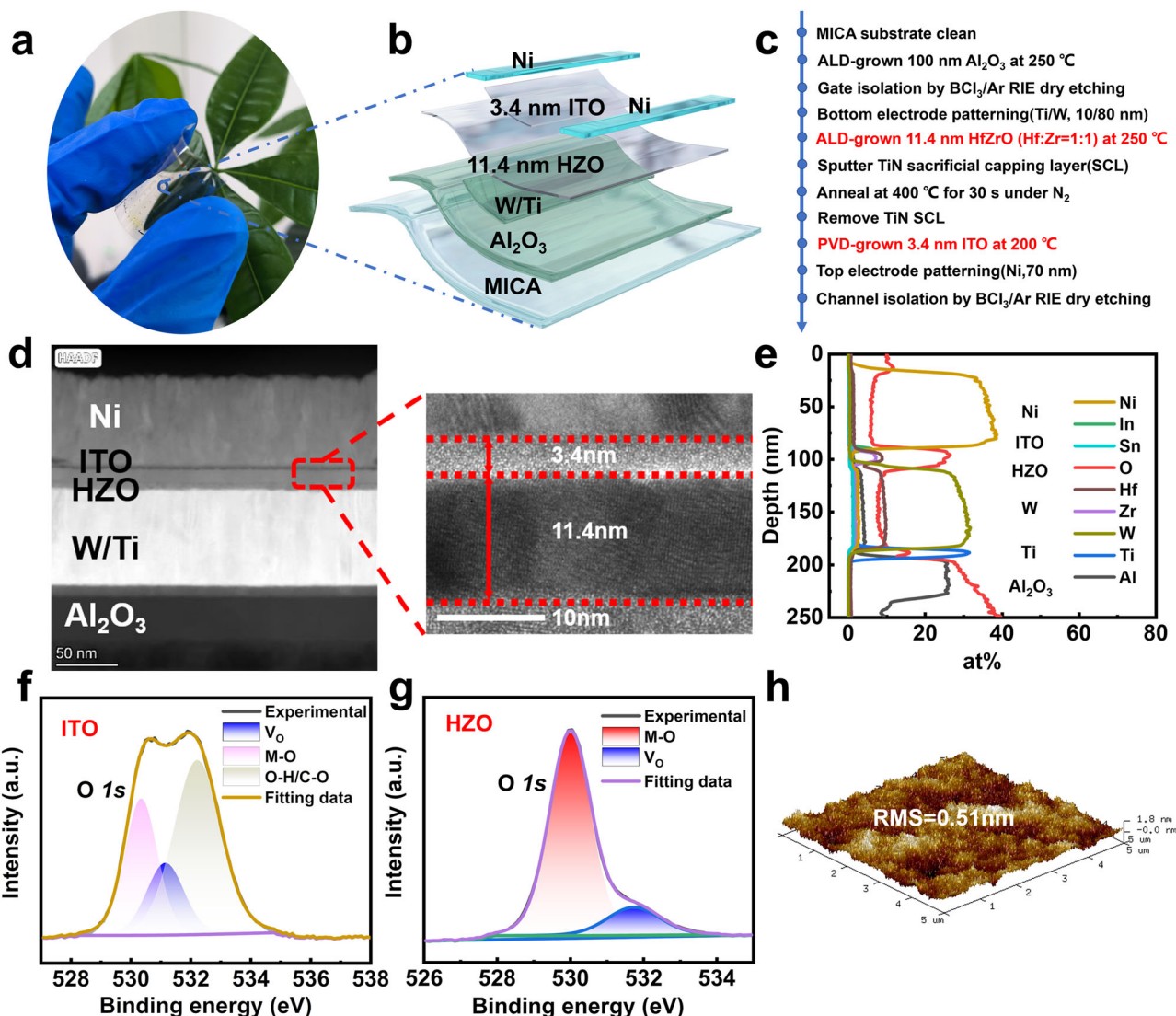

**Fig. 1 | Schematic and structural characterization of ITO FeFETs. a** Photograph of the device array in a bent state. The device is highly transparent and flexible. **b** Schematic diagram of BEOL-compatible flexible ITO FeFET memory devices. **c** The key experimental method flow of HZO-based FeFET fabrication process. The FE HZO is deposited at 250 °C with a cycle ratio of 1:1. **d** HRTEM image of the Ni/ITO/HZO/W/Ti/Al₂O₃ gate stack. HRTEM shows clear crystallization of the HZO material

and an ultra-thin 3.4 nm ITO channel. **e** Cross-sectional elemental analysis from EDX spectroscopy of ITO FeFETs. It can be clearly seen that the content of each element corresponds to the depth. The XPS spectra for O 1 s region of the **f** HZO film and **g** ITO channel. Oxygen vacancies account for 14.96% and 17.84% of all oxygen-related chemical bonds, respectively. **h** Atomic force microscope (AFM) image of the surface morphology of ITO film.

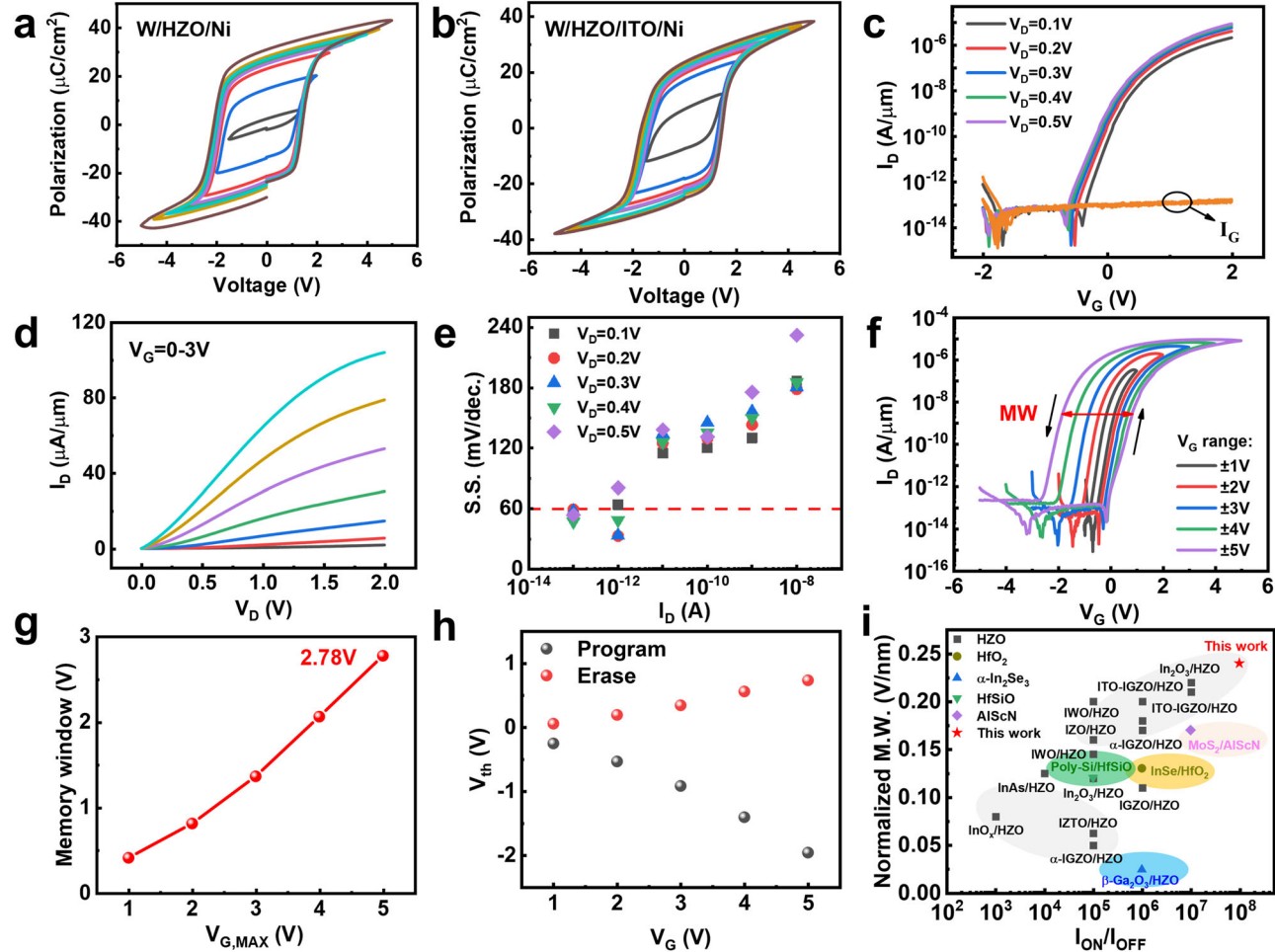

**Fig. 2 | Hysteresis and electrical properties of ITO FeFETs.** P-V loops of representative capacitors with **a** W/11.4 nm HZO /Ni and **b** W/11.4 nm HZO/3.4 nm ITO/Ni. The similar P-V characteristics suggest that the interface of FE and channel is excellent. **c** Measured $I_D$-$V_G$ of ITO FeFETs exhibiting near-ideal S.S. value, high $I_{ON}/I_{OFF}$ ratio of $10^8$, and sub-pA. gate/substrate leakage. **d** Measured $I_D$-$V_D$ curve, showing high $I_{on}$ of 105 μA/μm at $V_G$ = 3 V & $V_D$ = 0.1 V. **e** The S.S. distribution of ITO FeFETs devices, showing minimum S.S. of 33 mV/decade. **f** $I_D$-$V_G$ curves for ITO FeFETs. A stable FE type hysteresis with MW up to 2.78 V is available. **g** Memory window (MW) versus $V_{G, MAX}$ of ITO FeFETs at $V_D$ = 0.1 V. MW is calculated as $\Delta V_{th}$ in **f**. Record-high MW of 2.78 V at $V_{G, MAX}$ = 5 V. **h** Threshold voltages after erase and program versus $V_G$ Sweep range of ITO FeFETs at $V_D$ = 0.1 V. **i** Benchmarking of MW and $I_{ON}/I_{OFF}$ performance of FeFETs reported in this work (shown as a five-pointed star) versus recently reported FeFETs (shown as other symbols), where the MW is normalized with respect to the ferroelectric layer thickness.

large and stable memory window (-2.78 V). FeFET has high endurance of over $10^7$ cycles and good retention characteristics estimated to be over ten years. Under different bending conditions, the conductance modulation of the device has high stability and reliability. In addition, FeFET has excellent pulse cycles endurance (>5 × $10^5$) at a small bending radius (R = 5 mm). Additionally, we demonstrate the application of this device in neuromorphic computing. The excellent performance of flexible FeFET has greatly broadened the application of devices in the wearable field.

## Results
### Flexible ITO FeFET device structure
The device structure of the FeFETs is characterized. The optical microscope image of the ITO ferroelectric memory prepared on the flexible MICA substrate in the bent state is shown in Fig. 1a. The device is highly transparent and bendable. Figure 1b is an enlarged structural diagram of a single ITO FeFET. W/Ti and Ni are used as the bottom and top electrodes respectively, where the bottom electrode is prepared in the $Al_2O_3$ dielectric layer through a buried gate process. The specific preparation process of the device is shown in Fig. 1c. The thermal budget of the entire preparation process is within 400 °C BEOL compatibility. The micrograph of the device is shown in Supplementary

Fig. 1, displaying the shape of gate and source/drain with ITO channel. The channel length and width are 5 μm and 50 μm, respectively.

Figure 1d, e shows the High-resolution transmission electron microscope (HRTEM) and Energy dispersive x-ray spectroscopy (EDX) images of flexible ITO FeFETs capturing Ni/ ITO/ HZO/ W/ Ti/ $Al_2O_3$ stack. HRTEM shows clear crystallization of the HZO material and an ultra-thin 3.4 nm ITO channel. The cross-sectional TEM image of the FeFET and the EDS elemental map of each layer are shown in Supplementary Fig. 2. Figure 1f, g illustrates the X-ray photoelectron spectroscopy (XPS) for O *1s* region of the HZO and ITO films respectively. Oxygen vacancies account for 14.96% and 17.84% of all oxygen-related chemical bonds, respectively. The calculation method for oxygen vacancies is presented in Supplementary Fig. 3. The concentration of oxygen vacancies can be controllably altered by adjusting process parameters[23,37]. An appropriate concentration of oxygen vacancies is crucial for both ITO and HZO. The XPS spectra of ultra-thin 3.4 nm ITO fully shows that it is an excellent semiconductor[23]. The full XPS spectra of ITO film and HZO film are shown in Supplementary Fig. 4. Atomic force microscopy (AFM) was used to analyze the surface morphology of the ultra-thin ITO film. As can be seen from Fig. 1h, the root-mean-square (RMS) roughness of the ITO film is only 0.51 nm, which proves that through physical vapor deposition (PVD) can obtain a higher

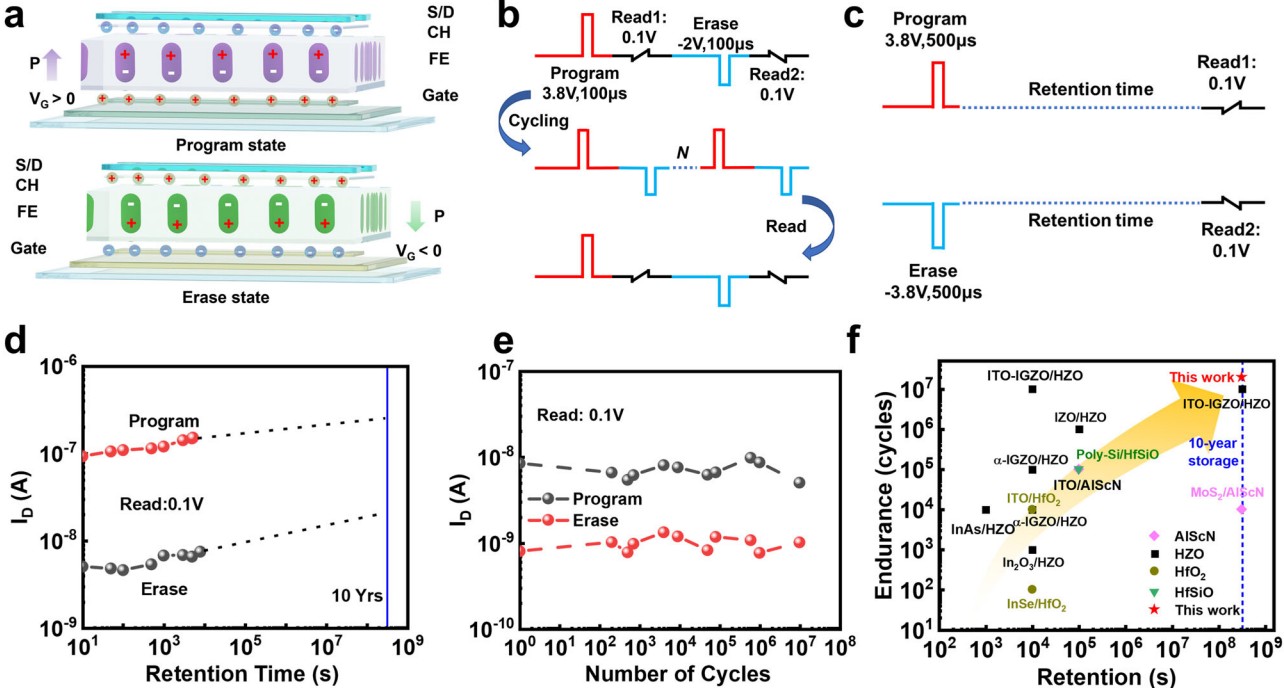

**Fig. 3 | Memory characteristics of ITO FeFETs. a** The charge distribution and polarization states of the ultra-thin ITO channel FeFET in programming mode (above) and erasing mode (below). **b** Pulse sequence for endurance test used. Pulse width of erase and program processes are $V_P$ (3.8 V, 100 μs) and $V_E$ (−2 V, 100 μs). **c** Pulse sequence for retention test used. Pulse width of both erase and program processes are 500 μs with amplitude of 3.8 V. **d** Retention characterization of the

ITO FeFETs. Optimized retention extrapolated as >10 years. at $V_D$ = 0.1 V. **e** Performance of the endurance property for the ITO FeFETs. High endurance exceeding $2 \times 10^7$ cycles. **f** Benchmarking the retention and endurance performance of FeFETs reported in this work (shown as five-pointed stars) against recently reported FeFETs (shown as other symbols).

quality ITO channel and ensure the performance of the device. The AFM surface analysis of the HZO film is shown in Supplementary Fig. 5, and the RMS is only 1.27 nm.

### Device performance of flexible ITO FeFETs

The P-V loops in different voltage of metal/insulator/metal (MIM) capacitors with the structures (a) W/11.4 nm HZO/Ni and (b) W/11.4 nm HZO/3.4 nm ITO/ Ni are shown in Fig. 2a, b. The similar P-V characteristics suggest that the interface of FE and channel is excellent. Different annealing temperatures and electrodes have a significant impact on the ferroelectric properties of HZO. Low-temperature annealing leads to a decline in ferroelectric performance[38,] while excessively high annealing temperatures result in larger leakage currents[39]. Since TiN is wet-etched easily and can endow the ferroelectric thin films with larger remnant polarization (Pr) and lower leakage current[40], TiN was chosen as the electrode for HZO annealing. Figure 2c displays the measured $I_D$-$V_G$ of ITO FeFETs which exhibits a near-ideal S.S. value, high $I_{ON}/I_{OFF}$ ratio of $10^8$, and sub-pA gate/substrate leakage. Magnified images of the transistor gate current and the leakage current of the HZO gate stack at different applied voltages are shown in Supplementary Fig. 6. The low level of leakage current ensures the reliability and stability of the device. Figure 2d shows the $I_D$-$V_D$ curves, showing high $I_{ON}$ of 105 μA/μm at $V_G$ = 3 V and $V_D$ = 0.1 V. In the small $V_D$ range of 0–0.5 V, $I_D$ and $V_G$ exhibit linearity, demonstrating a good ohmic contact between S/D and channel (Supplementary Fig. 7). The S.S. distribution suggests a good ferroelectric switch in Fig. 2e. At room temperature, S.S. (33 mV/decade) of sub-60 was obtained.

Figure 2f demonstrates the $I_D$-$V_G$ curves at different $V_G$ sweep ranges for ITO FeFETs with a stable FE type hysteresis. Memory window (MW) versus $V_G$ ranges is shown in Fig. 2g, and at $V_{G, MAX}$ = 5 V and $V_D$ = 0.1 V, a record-high MW of 2.78 V is available. Figure 2h shows the threshold voltages ($V_{th}$) after erase and program versus $V_G$, ± 2 V is

enough to program and erase with a large MW. The distribution of key performance indicators (KPIs) for 30 devices is presented in Supplementary Fig. 8, including critical performance information such as the on/off ratio, S.S., MW, and $I_{ON}$. The overall performance of the devices is favorable, which can be attributed to mature device fabrication processes and stable experimental conditions. Figure 2i benchmarks the MW and $I_{ON}/I_{OFF}$ performance of the FeFET reported in this work (shown as a five-pointed star) against those reported in the last three years (shown as other symbols), with the MW is normalized relative to the ferroelectric layer thickness[19,41–57]. Supplementary Table 1 summarizes the key parameters of the flexible ITO FeFET in this study compared with the control group. This device successfully implements ultra-thin ITO channel high-performance BEOL-compatible ferroelectric memory on a flexible substrate, and has obvious advantages over other FeFETs, including S.S. as small as 33 mV/decade and $I_{ON}/I_{OFF}$ ratio exceeding $10^8$ and record high normalized MW of 0.24 V/nm.

The operational mechanism of FeFET is shown in Fig. 3a. Ferroelectric polarization switches can modulate the $V_{th}$ of the device by reversing the polarity of the polarization-bound charges, thereby achieving a counterclockwise $I_D$-$V_G$ curve. When a positive gate voltage is applied, the ferroelectric polarization is oriented upwards, resulting in the accumulation of charges in the channel, corresponding to the programming state. Conversely, when a negative gate voltage is applied, the ferroelectric polarization is reversed, leading to depletion of charge carriers in the channel, corresponding to the erasing state. Figure 3b, c shows pulse sequence for endurance and retention test. For endurance measurement, the pulse width of erase and program processes are $V_P$ (+3.8 V,100 μs) and $V_E$ (−2 V,100 μs). For retention measurement, the pulse width of both erase and program processes are 500 μs with amplitude of 3.8 V. Figure 3d shows the retention characteristics of the ITO FeFETs. Optimized retention extrapolated as >10 years at $V_D$ = 0.1 V. Figure 3e displays the great performance of the

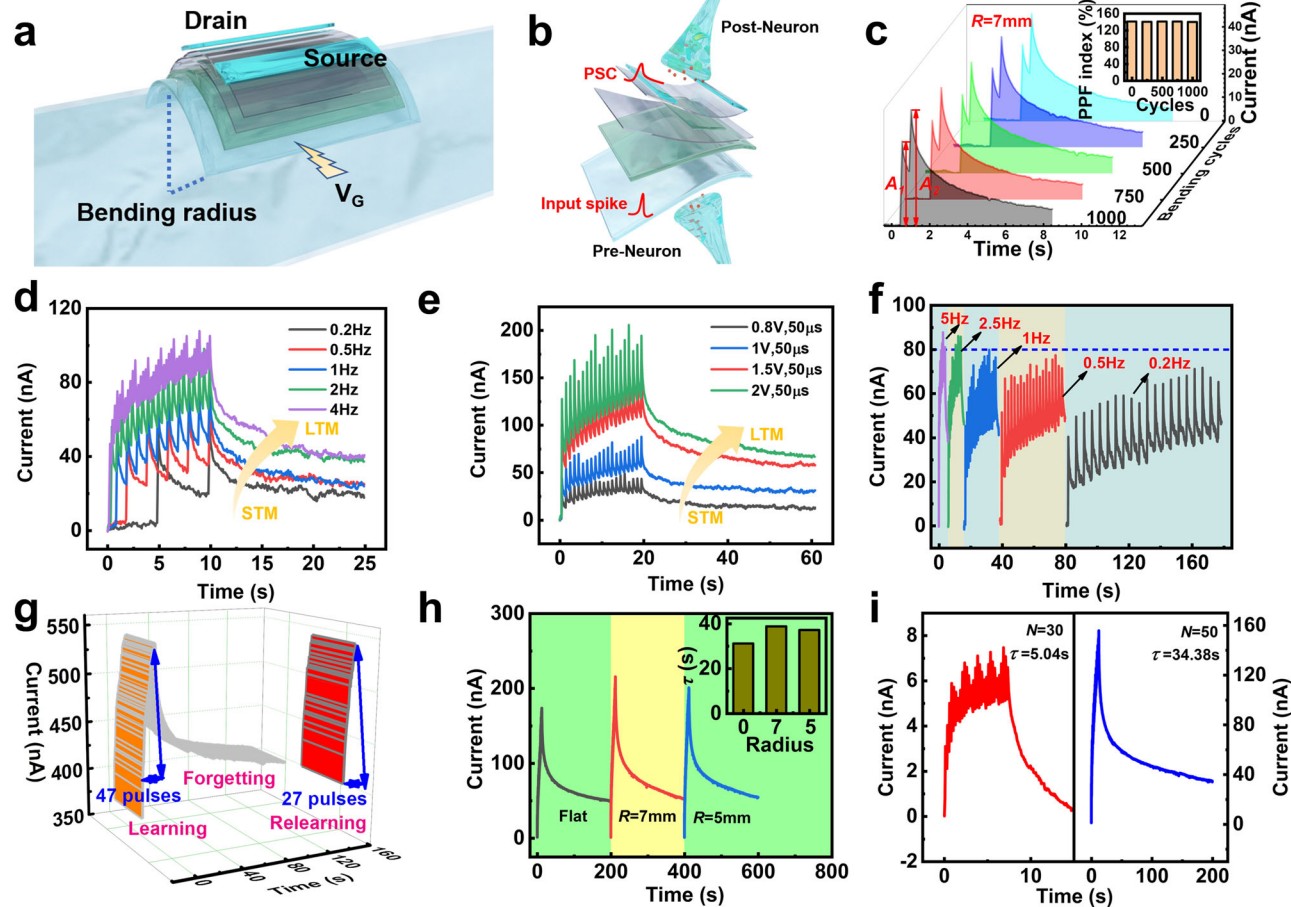

**Fig. 4 | Non-volatile synaptic behavior of flexible ITO FeFETs. a** Schematic diagram of device flexibility testing. **b** The information transmission at biological synapses and corresponding artificial synaptic devices. **c** PPF under different bending cycles caused by paired pulses with a time interval of 500 ms. The inset is the PPF index extracted from the curve, constant at 141%. The transition from STM to LTM is achieved by changing the applied **d** pulse frequency and **e** pulse amplitude. **f** High-pass filtering characteristics of ITO FeFETs. **g** Learning behavior of ITO FeFETs. After 130 s of forgetting process, the FeFETs can reach the initial current level after only 27 same pulses revealing excellent memory retention. **h** The learning-forgetting process of ITO FeFETs under different bending radii of 5 mm and 7 mm. The inset presents the dependence of relaxation time on radius. **i** Relaxation time constant ($\tau$) as a function of applied pulse parameters.

ITO FeFETs endurance property and a high endurance exceeding $2 \times 10^7$ cycles is achieved. Figure 3f benchmarks the retention and endurance performance of FeFETs reported in this work (shown as five-pointed stars) against those reported in the past three years (shown as other symbols)[19,41,47,48,51–59]. A more detailed parameter comparison is shown in Supplementary Table 2. Compared with other FeFETs, the flexible ultrathin ITO channel ferroelectric memory has longer data retention time and better endurance characteristics.

**Experimental characterization of synaptic properties of flexible ITO FeFETs**

The neuromorphic computing properties of the ITO FeFETs under different bending test conditions are studied. Figure 4a is a schematic diagram of the test of a flexible device under a certain bending radius. The highly transparent and flexible devices are bent on the surface of a cylindrical with different radii ($R = 7$ mm or $R = 5$ mm). The real-time test photos of the FeFETs on the ultrathin MICA being characterized under the folded state in Supplementary Fig. 9. The information transfer of biological synapses and corresponding artificial synaptic devices is shown in Fig. 4b. Excitatory or inhibitory stimuli are transmitted from the presynaptic neuron to the postsynaptic neuron across the synaptic cleft. In artificial synapse device, the gate and source/drain electrodes correspond to presynaptic and postsynaptic neurons, respectively. Synaptic plasticity corresponds to the postsynaptic current (PSC), the conductance level of the device.

Paired pulse facilitation (PPF) is an important manifestation of short-term synaptic plasticity. Triggered by pulses (3 V, 200 μs) with a time interval of 500 ms, paired-pulse facilitation (PPF) under different bending cycles of $R = 7$ mm is displayed in Fig. 4c. The inset is the PPF index ($A_2/A_1$) extracted from the curve, constant at 141% which illustrates that it maintains good state after bending 1000 cycles. In addition to short-term synaptic plasticity, long-term synaptic plasticity is crucial for organisms' behaviors such as learning and memory[60,61]. The transition from short-term memory (STM) to long-term memory (LTM) can be achieved in ITO artificial synaptic devices by changing the applied pulse parameters. When the parameters of a single applied pulse are fixed (1 V, 50 μs), as the applied pulse frequency increases from 0.2 Hz to 4 Hz in Fig. 4d, the level of PSC gradually increases and acquires a slower decay rate. Similarly, when applying 20 consecutive pulses with a fixed time interval of 1 s and a pulse width of 50 μs, as the pulse amplitude increases from 0.8 V to 2 V, the promotion of memory level is achieved. (Fig. 4e).

In biological nervous systems, synapses with low neurotransmitter release probability can function as high-pass filters, which is crucial for organisms to process redundant information[62]. Figure 4f shows the simulation of high-pass filtering characteristics. 20 continuous pulses (1 V, 50 μs) are applied to the ITO FeFET at different frequencies. If the cut-off current (blue dotted line) is set to 80 nA, only the frequency is higher than 1 Hz signals can be transmitted. In addition, we simulated the learning behavior of organisms, and the results

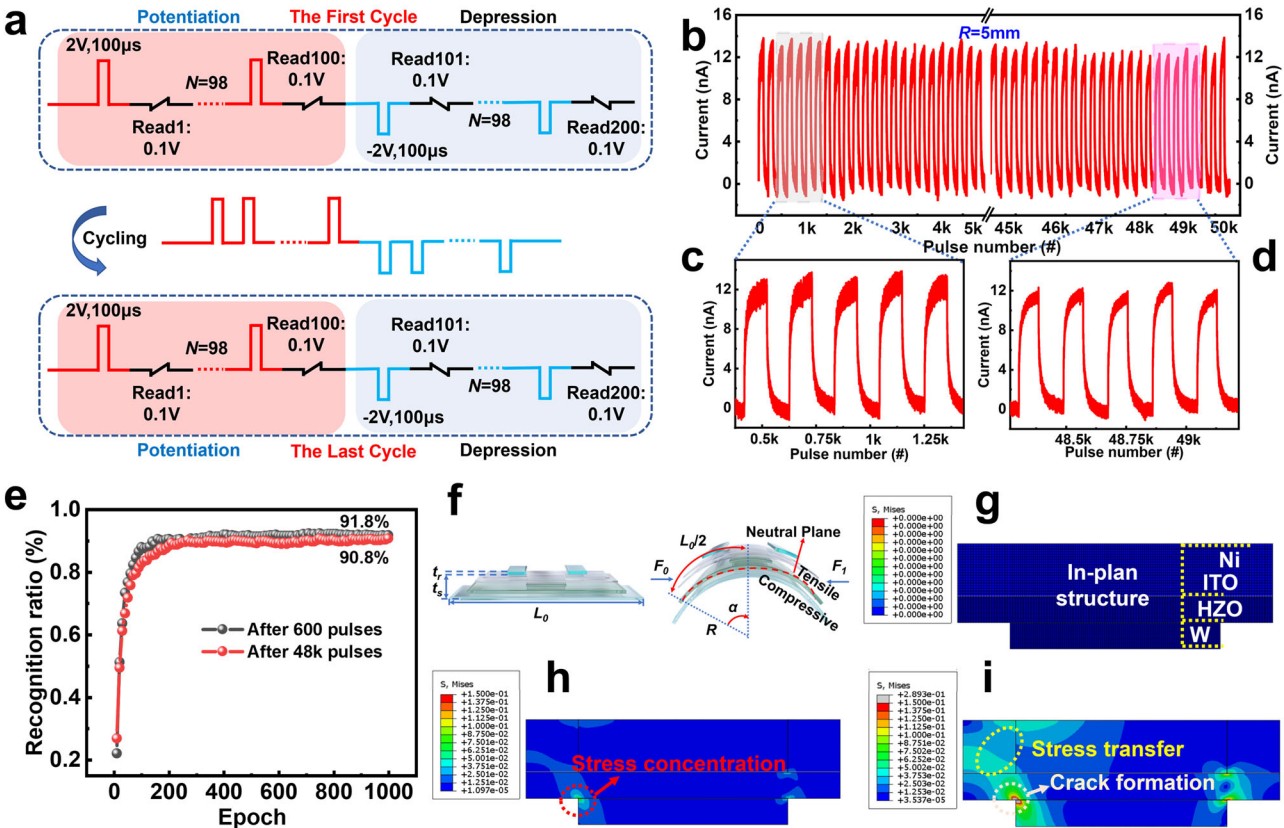

**Fig. 5 | Bending Reliability of ITO FeFETs. a** Pulse sequence for Potentiation-depression cycling stress tests used. One cycle was 100 potentiating pulses and the subsequent 100 depressing pulses. All the currents were measured with an applied bias of −0.1 V on the postsynaptic neuron. **b** Repetitive LTP and LTD operations in the folded FeFETs during 50 K spikes of presynaptic pulses. **c**, **d** Show the LTP and LTD during the initial and final 5 cycles, respectively. **e** MNIST pattern recognition accuracy was evaluated by backpropagation (BP) algorithm. A high recognition accuracy close to the ideal was obtained in ITO FeFETs after applying 600 pulses and 48 K pulses. **f** Schematic diagram of the device's bending strain. **g–i** Stress variation of the device during the bending process. The stress distribution on different functional layers was obtained through FEA simulation as the bending radius changed.

are shown in Fig. 4g. First, 47 pulses (3 v, 200 μs) are applied to bring the organism's memory to a higher level. After 130 s of forgetting process, the ITO FeFETs can reach the initial current level after only 27 same pulses, which proves its excellent memory retention ability.

In addition, ITO FeFETs under different bending radii were used to simulate the learning-forgetting behavior of organisms (Fig. 4h). First, 50 consecutive pulses (3 V, 200 μs) are applied to devices with different bending radii (flat, $R = 7$ mm, $R = 5$ mm) to make the PSC reach a high level, followed by a natural forgetting process of up to 200 s. Memory retention was further quantified using the Kohlrausch equation[63,64]:

$$I_t = I_e + Ae^{\frac{-t}{\tau}}, A = I_0 - I_e \qquad (1)$$

where $I_e$ is the current at the final steady state, $t$ is time, and $\tau$ is the relaxation time constant. The $\tau$ obtained by the Kohlrausch equation is shown in the inset of Fig. 4h. Under three bending radii, the conductance level of the device is continuously and stably modulated, and the $\tau$ is similar. When the number of pulses ($N = 30/50$) increases and the pulse parameters (2 V, 100 μs or 3 V, 200 μs) increase, the $\tau$ increases exponentially, which also corresponds to the improvement of memory level and the formation of LTM (Fig. 4i).

## Bending reliability of flexible ITO FeFETs

Next-generation flexible and wearable neuromorphic computing devices must be able to maintain stable synaptic properties under small bending radii and have continuous and uniform conductance

switching behavior. Figure 5a demonstrates the used pulse sequence for potentiation-depression cycling stress tests and one cycle consists of 100 potentiating pulses and the subsequent 100 depressing pulses. All the currents are measured with an applied bias of −0.1 V on the postsynaptic neuron. The repetitive long-term potentiation/depression (LTP/LTD) operations in the folded devices during $5 \times 10^5$ spikes of presynaptic pulses (±2 V, 100 μs) are measured in Fig. 5b. The initial and last 5 cycles LTP and LTD operations reveal the pulse cycling robustness of the FeFETs under bending conditions (Fig. 5c, d). Additionally, we have investigated the impact of different bending radii and bending cycles on the response of the HZO gate stack and the $I_{ON}/I_{OFF}$ ratio, endurance, and other performances of FeFETs, as shown in Supplementary Figs. 10 and 11. The device performance remains stable under different application scenarios.

## Artificial neural networks based on ITO FeFET synaptic devices

We constructed an artificial neural network (ANN) using ITO FeFETs to recognize handwritten digits in the MNIST database (Supplementary Fig. 12). The network consists of an input layer (784 neurons), a hidden layer (64 neurons), and an output layer (10 neurons)[65]. The process of training and recognizing digital images using ANN[65,66], including signal forward propagation and error backpropagation, is depicted in Supplementary Fig. 13. The relationship between network recognition accuracy and factors such as the number of neurons in the hidden layer and the initial values of weights $V$ and $W$, which significantly influence the accuracy and efficiency of ANN classification, is shown in Supplementary Fig. 14. The correlation between recognition accuracy and the

proportion of noise pixels is demonstrated in Supplementary Fig. 15, illustrating the robust fault tolerance of the constructed ANN. The device conductance levels after 600 and 48,000 pulse operations were extracted for network training. The average confusion matrix change during the pattern training epochs is shown in Supplementary Fig. 16. As the training process proceeds, the inferred output becomes consistent with the desired output. After 1000 iterations of training based on the backpropagation (BP) algorithm, the recognition rates of the two groups of flexible ferroelectric neural networks reached 91.8% and 90.8%, respectively (Fig. 5e). This negligible difference is crucial for wearable platforms to be used for more complex neural processing to realize truly wearable smart electronics.

In order to comprehend the impact of stress distribution on the degradation of device performance, finite element analysis (FEA) was conducted using ABAQUS software. Figure 5f illustrates the schematic of bending strain for the two-point bending operation, where stress is applied from both sides of the device. A two-dimensional model for the device structure was established, as depicted in Supplementary Fig. 17, with the material parameters outlined in Supplementary Table 3. The model assumes perfect bonding at the interfaces between different layers. In the simulation, the folding process was simulated by incrementally increasing the displacement loading (DL). In the initial state (Fig. 5g), no apparent structural defects were observed in the device. As shown in Fig. 5h, with the gradual reduction of the bending radius, the internal stress of the device increased, leading to the formation of stress concentration points. With the continued application of external pressure, the stress concentration in the device readily induced crack formation and propagation within the device, ultimately resulting in a deterioration of device performance or even device failure (Fig. 5i). Analyzing the impact of displacement loading on the device's lifespan contributes to understanding stress propagation paths and exploring device failure mechanisms. This, in turn, facilitates the enhancement of the reliability of flexible devices subjected to bending stress.

## Discussion

In summary, addressing the challenges of achieving high-performance ferroelectric memory on flexible substrates, based on ultra-thin (3.4 nm) ITO channel and ferroelectric material HZO, a BEOL-compatible high-performance ITO FeFET was prepared on a flexible MICA substrate with a thermal budget below 400 °C. Since the PVD ITO channel exhibits a low thermal budget of 200 °C, highly controllable thickness, wafer-level uniformity and conformality, and ultra-high electron mobility, the prepared FeFET has a minimum S.S. of 33 mv/decade, an $I_{ON}/I_{OFF}$ ratio exceeding $10^8$, and an excellent large MW of 2.78 V. In terms of memory characteristics, the endurance of ITO FeFETs >$2 \times 10^7$ cycles, and the data retention >10 years. The above parameters are at the leading level among reported FeFETs. In addition, the prepared ITO FeFET is transparent and bendable. The device successfully simulates a variety of biological synaptic behaviors under different bending radii, and its performance remains stable after multiple bending cycles. The most important is that when $5 \times 10^5$ pulses are continuously applied to the device with a bending radius of 5 mm, the conductance modulation behavior of the device is uniform and stable. This demonstrates the device's bending reliability and excellent high pulse cycle durability. We believe that continuous efforts in this innovative approach are expected to pave the way for future electronic devices, promoting the application of ferroelectric memories in wearable, foldable electronics.

## Methods

### Device fabrication

Figure 1c illustrates the key experimental process. After solvent clean of the MICA substrate, 100 nm $Al_2O_3$ was grown at 250 °C by atom layer deposition (ALD) using $(CH_3)_3Al$ (TMA) and $H_2O$ as precursors. Gate isolation was performed by dry etching by $BCl_3/Ar$. Then the bottom electrode was patterned by lithography and deposited 10 nm Ti/80 nm W by sputtering. The FE 11.4 nm HZO was grown by ALD at 250 °C with a cycle ratio of 1:1. 80 nm TiN was sputtered as a sacrificial capping layer (SCL) and rapid thermal annealing (RTA) at 400 °C for 30 s in $N_2$. The TiN SCL was removed by a mixture of ammonia, hydrogen peroxide, and water ($NH_3$: $H_2O_2$: $H_2O$ = 1:1:5). The 3.4 nm ITO was deposited by sputtering at 200 °C under an oxygen ratio of 15% ([O]/[O]+[Ar] = 15%). The top electrode 70 nm sputtered Ni was patterned by bi-layer photoresist lithography as Source/ Drain (S/D). Channel area was defined by $BCl_3/Ar$ dry etching. The channel length and width were 5 and 50 μm. The thermal budget of the entire fabrication process is below 400 °C, so the device performance is expected to remain unchanged under back-end-of-line (BEOL) conditions.

### Device characterization

The basic electrical characteristics and synaptic characteristics of ITO FeFETs were tested under dark conditions and room temperature by Agilent B1500A semiconductor parameter analyzer. Pulse signals with different parameters (amplitude, width) are applied to the device through an arbitrary function generator (KEYSIGHT B1525A). The ferroelectric polarization hysteresis loop of the device was measured with a precision LC analyzer (Radiant Technologies Inc.). In the cross-sectional TEM test of the device, FIB precision sample preparation (FEI Helios) was first performed, followed by TEM analysis (FEI Talos F200X). X-ray photoelectron spectroscopy (XPS) was obtained using a Thermo ESCALAB 250Xi XPS system. Atomic force microscopy (AFM; Dimension Edge, Bruker) was used to analyze the surface morphology of ITO and HZO films.

## Data availability

The source data that support the findings of this study are available in figshare with the identifier [https://doi.org/10.6084/m9.figshare.25374901].

## Code availability

Code from this study is available from the corresponding author upon request.

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

## Acknowledgements

This work was supported by the National Key R&D Program of China (Grant 2021YFA1202600), the NSFC (Grants 92064009, 22175042), Science and Technology Commission of Shanghai Municipality (22501100900), China Postdoctoral Science Foundation (Grant 2022TQ0068, BX2021070, 2021M700026), and the Zhejiang Lab's International Talent Fund for Young Professionals.

## Author contributions

Q.L., S.W., and L.C. initiated the research. Q.L., S.W., and J.Y. prepared and characterized the ITO and HZO. Q.L. and X.H. designed the ferroelectric transistors. Q.L. and S.W. fabricated the ferroelectric synaptic transistor. Q.L., Q.S., and Y.Y. conducted the electrical measurements. Q.L. and Z.L. conducts flexible measurement and FEA simulation. Q.L., Q.S., L.C., and D.W.Z. analyzed the data. Q.L. and D.W.Z. wrote the manuscript with input from all the other authors. T.W., Y.L., and J.M. provided support during the manuscript revision process. All authors discussed the results and commented on the manuscript.

## Competing interests

The authors declare no competing interests.
