## [Peer Review File · Nature Communications]

REVIEWER COMMENTS

Reviewer #1 (Remarks to the Author):

Please find in the attached document.

Reviewer #2 (Remarks to the Author):

The paper reports outstanding results of implemented FeFET on transparent and flexible substrates.

We recommend major revision in order to complete the full picture and also clearing minor typos.

The paper states excellent performance values of implemented FeFET. However, background information as well as micrograph about device geometry is missing and would be helpful to understand and compare the device performance.

Further it would be worth to discuss impact of pulse width on program- / erase performance including discussion about potential impact of trapping on the switching performance. Based on the subloop switching behavior and matching with the PV characteristic no saturation in Fig 2f is visible which points to a high risk of trapping as prevailing switching mechanism rather than FE switching. Also I think there is a mismatch between Fig 2f and 2h

In the neuromorphic section I recommend a better differentiation between experimental and simulated results.

Reviewer #3 (Remarks to the Author):

In the work "High-performance ferroelectric field-effect transistors with ultra-thin indium tin oxide channels for flexible and transparent electronics" the authors demonstrate for the first time the integration of an HZO based oxide-channel FET on a flexible substrate of mica with an annealing at 400°C.

Even though, this is an interesting further progress with novel results on the FET properties, particularly also its synaptic properties, there has been extensive work already in the field demonstrating:

- HZO annealing for BEOL compatible integration with HZO annealing at or below 400°C with a multitude of FRAM as well as FeMFET demonstrations
- the integration of HZO on IGZO channel with such a temperature profile compatible to BEOL conditions has been demonstrated on Si substrates
- the bending resistance of HZO based RRAM cells has been demonstrated on mica substrates before
- synaptic behavior in similar ferroelectric FETs

Hence, some clarifications could be helpful to highlight the key improvement of the manuscript.

Furthermore, I would be interested what is the expectation of the authors on the influence of the in-plane stress on the ferroelectric properties of the material? There have been reports, that the in-plane stress has a large influence in the formation of ferroelectric/antiferroelectric properties upon crystallization.

More measurement data should be shown concerning the basic ferroelectric FET properties upon bending cycles. So far only the synaptic properties are shown.

The SS slope significantly below 60mV/dec does not appear to be consistently apparent. Can the authors support the evaluation with further measurement?

The article titled " High-performance ferroelectric field-effect transistors with ultra- thin indium tin oxide channels for flexible and transparent electronics" by Q. Li et al. reported an ultrathin Indium-Tin-Oxide (ITO) and $\text{Hf}_{0.5}\text{Zr}_{0.5}\text{O}_2$ (HZO) based low-thermal budget FeFET on flexible substrates for memory and neuromorphic computing circuits. While the reported device performance shows promising numbers, the concept and performance of the device presented here is not really pathbreaking. Therefore, I would like to suggest modifications of the current manuscript in order to accept its publication in Nature Communications. Following are some concerns and questions that I have and some suggestions that I would like to propose.

1. In Introduction, it is stated that "Up to now, most FeFETs are fabricated on rigid substrates, and the few reported ferroelectric devices fabricated on flexible substrates are difficult to obtain excellent performance due to the limitation of process temperature". This statement does not reflect the landscape properly. For instance, many P(VDF-TrFE) based systems showed excellent performance. These works should be mentioned, and the authors should point out the additional advantage that HZO based systems are supposed to bring in.
2. "And they also suffer from lower durability, typically around 10^3 cycles." This statement is not correct. In flexible FeFETs, even 10^8 cycles of operation has been demonstrated. For instance, please check the state-of-the-art numbers from several different flexible FeFET structures from this work: <https://onlinelibrary.wiley.com/doi/full/10.1002/aisy.202100175>.
3. With respect to Fig. 1, it is written that "Oxygen vacancies account for 14.96% and 17.84% of all oxygen-related chemical bonds, respectively." How is this calculated? Can the authors explain the process in the SI? Is the oxygen vacancy contribution dependent on the stack? Can the oxygen vacancy concentration be modified controllably with fab process parameters or stack designs? What would be the effect on performance both for ITO and HZO?
4. What are the leakage current density of the devices? The leakage current of the HZO gate stack and the transistor gate current should be plotted.
5. A wafer-scale performance variation mapping would be important. At least a distribution of KPIs for more than 25-30 devices can provide an accurate picture. Prediction based on single device performance, is often over-optimistic.
6. HZO gate stack response and FeFET performance under different bending radii would be important to have. Due to piezoelectric nature of the ferroelectric thin films, additional charge generation due to bending (and hence deformation of the FE) is expected. It would be important to know how they affect device properties like V_c , MW, I On/off, endurance.
7. The training protocol is not clear from the discussion. most likely, it is based on single device performance extrapolated to form a crossbar? What would be the configuration and applied pulse protocols for the training? How does the size of the network and task complexity affect the classification accuracy? How could cycle-to-cycle and device-to-device performance variation could affect the network performance?
8. Multiple synaptic plasticity with varied time constants is mentioned in the article. Can a network based on these FeFETs able to perform spatio-temporally varying dataset?

Response Letter

Response to Reviewer 1's Comments:

Comment: The article titled “High-performance ferroelectric field-effect transistors with ultra- thin indium tin oxide channels for flexible and transparent electronics” by Q. Li et al. reported an ultrathin Indium Tin-Oxide (ITO) and $\text{Hf}_{0.5}\text{Zr}_{0.5}\text{O}_2$ (HZO) based low-thermal budget FeFET on flexible substrates for memory and neuromorphic computing circuits. While the reported device performance shows promising numbers, the concept and performance of the device presented here is not really pathbreaking. Therefore, I would like to suggest modifications of the current manuscript in order to accept its publication in Nature Communications. Following are some concerns and questions that I have and some suggestions that I would like to propose.

Response: We sincerely appreciate the reviewer's positive and encouraging comments on our manuscript. We have provided corresponding responses to the reviewer's questions below and made necessary modifications in the revised manuscript. We hope our revisions could meet the quality require of Nature Communications. We thank the reviewer again for the very good suggestions.

Comment 1: In Introduction, it is stated that “Up to now, most FeFETs are fabricated on rigid substrates, and the few reported ferroelectric devices fabricated on flexible substrates are difficult to obtain excellent performance due to the limitation of process temperature”. This statement does not reflect the landscape properly. For instance, many P(VDF-TrFE) based systems showed excellent performance. These works should be mentioned, and the authors should point out the additional advantage that HZO based systems are supposed to bring in.

Response 1: Thank you for your comment. We apologize if the initial statement in the introduction did not fully reflect the breadth of research in this field. We have revised the introduction to include the achievements of P(VDF-TrFE) based systems and the additional advantages that HZO-based systems offer. The revised parts are highlighted

in the revised manuscript. Furthermore, the manuscript has been supplemented with numerous critical references to emphasize the significance of the work and the main challenges, which helps to strengthen the introduction part. The rephrased parts and added discussions are as follows:

" Until now, FeFETs based on rigid substrates have exhibited superior performance. Although ferroelectric devices fabricated on flexible substrates, such as systems based on polyvinylidene fluoride (PVDF) and its copolymer polyvinylidene fluoride-trifluoroethylene (PVDF-TrFE), have also demonstrated excellent capabilities, proven to exceed 10^8 cycles of endurance⁸⁻¹⁰, they are confronted with limitations of film thickness and driving voltage¹¹⁻¹⁴."

" P(VDF-TrFE) is cost-effective, easy to fabricate via sol-gel processing, and requires a low processing temperature, making it a promising ferroelectric material for FeFETs and synaptic devices^{11,24-26}. However, P(VDF-TrFE) films are typically thick (over 100nm) and necessitate higher driving voltages for polarization switching and saturation¹¹⁻¹⁴. In contrast, thin Zr-doped HfO₂ (HZO) exhibits outstanding ferroelectric performance at low driving voltages even at a thickness of merely 5nm²⁷⁻²⁹, which is crucial for scalable integrated circuits incorporating ultra-thin oxide semiconductors."

8 Majumdar, S. Back-end CMOS compatible and flexible ferroelectric memories for neuromorphic computing and adaptive sensing. *Advanced Intelligent Systems* **4**, 2100175 (2022).

9 Li, Q.-X. *et al.* Ferroelectric artificial synapse for neuromorphic computing and flexible applications. *Fundamental Research* **3**, 960-966 (2023).

10 Dang, Z., Guo, F., Wu, Z., Jin, K. & Hao, J. Interface Engineering and Device Applications of 2D Ultrathin Film/Ferroelectric Copolymer P (VDF-TrFE). *Advanced Physics Research* **2**, 2200038 (2023).

11 Dang, Z. *et al.* Black Phosphorus/Ferroelectric P (VDF-TrFE) Field-Effect Transistors with High Mobility for Energy-Efficient Artificial Synapse in High-Accuracy Neuromorphic Computing. *Nano Letters* (2023).

12 Rahi, S., Raghuwanshi, V., Konwar, G. & Tiwari, S. P. High-Performance Flexible Solution-Processed Organic Nonvolatile Memory Transistors. *IEEE Transactions on Electron Devices* (2023).

13 Lee, K. H. *et al.* High-mobility nonvolatile memory thin-film transistors with a ferroelectric polymer interfacing ZnO and pentacene channels. *Advanced Materials* **21**, 4287-4291 (2009).

14 Lee, K.-J., Yang, T.-Y., Chou, D.-W. & Wang, Y.-H. Hybrid Ferroelectric P (VDF-TrFE)/BZT Insulators for Pentacene-Based Nonvolatile Memory Applications. *IEEE Electron Device Letters* **43**, 1463-1466 (2022).

24 Shen, C.-K., Chaurasiya, R., Chen, K.-T. & Chen, J.-S. Synaptic Emulation via Ferroelectric P (VDF-TrFE) Reinforced Charge Trapping/Detrapping in Zinc–Tin Oxide Transistor. *ACS Applied Materials & Interfaces* **14**, 16939-16948 (2022).

25 Lee, Y. T. *et al.* Nonvolatile ferroelectric memory circuit using black phosphorus nanosheet-based field-effect transistors with P (VDF-TrFE) polymer. *Acs Nano* **9**, 10394-10401 (2015).

26 Chu, F. J., Chen, Y. C., Shih, L. C., Mao, S. C. & Chen, J. S. Reconfigurable Physical Reservoir Enabled by Polarization of Ferroelectric Polymer P (VDF–TrFE) and Interface Charge-Trapping/Detrapping in Dual-Gate IGZO Transistor. *Advanced Functional Materials*, 2310951 (2023).

27 Ali, T. *et al.* in *65th IEEE Annual International Electron Devices Meeting (IEDM)*. (2019).

28 Kim, S. J. *et al.* Low-voltage operation and high endurance of 5-nm ferroelectric Hf_{0.5}Zr_{0.5}O₂ capacitors. *Applied Physics Letters* **113** (2018).

29 Lyu, X. *Polarization and Switching Dynamics Study of Ferroelectric Hafnium Zirconium Oxide for Feram and Fefet Applications*. (2023).

The revised parts have been highlighted in the manuscript.

Corresponding change in manuscript: Yes

Location of Change:

- On Page 3,

“Until now, FeFETs based on rigid substrates have exhibited superior performance. Although ferroelectric devices fabricated on flexible substrates, such as systems based on polyvinylidene fluoride (PVDF) and its copolymer polyvinylidene fluoride-trifluoroethylene (PVDF-TrFE), have also demonstrated excellent capabilities, proven to exceed 10^8 cycles of endurance⁸⁻¹⁰, they are confronted with limitations of film thickness and driving voltage¹¹⁻¹⁴.”

8 Majumdar, S. Back - end CMOS compatible and flexible ferroelectric memories for neuromorphic computing and adaptive sensing. *Advanced Intelligent Systems* **4**, 2100175 (2022).

9 Li, Q.-X. *et al.* Ferroelectric artificial synapse for neuromorphic computing and flexible applications. *Fundamental Research* **3**, 960-966 (2023).

10 Dang, Z., Guo, F., Wu, Z., Jin, K. & Hao, J. Interface Engineering and Device Applications of 2D Ultrathin Film/Ferroelectric Copolymer P (VDF-TrFE). *Advanced Physics Research* **2**, 2200038 (2023).

11 Dang, Z. *et al.* Black Phosphorus/Ferroelectric P (VDF-TrFE) Field-Effect Transistors with High Mobility for Energy-Efficient Artificial Synapse in High-Accuracy Neuromorphic Computing. *Nano Letters* (2023).

12 Rahi, S., Raghuwanshi, V., Konwar, G. & Tiwari, S. P. High-Performance Flexible Solution-Processed Organic Nonvolatile Memory Transistors. *IEEE Transactions on Electron Devices* (2023).

13 Lee, K. H. *et al.* High-mobility nonvolatile memory thin-film transistors with a ferroelectric polymer interfacing ZnO and pentacene channels. *Advanced Materials* **21**, 4287-4291 (2009).

14 Lee, K.-J., Yang, T.-Y., Chou, D.-W. & Wang, Y.-H. Hybrid Ferroelectric P (VDF-TrFE)/BZT Insulators for Pentacene-Based Nonvolatile Memory Applications. *IEEE Electron Device Letters* **43**, 1463-1466 (2022).

“P(VDF-TrFE) is cost-effective, easy to fabricate via sol-gel processing, and requires a low processing temperature, making it a promising ferroelectric material for FeFETs and synaptic devices^{11,24-26}. However, P(VDF-TrFE) films are typically thick (over 100nm) and necessitate higher driving voltages for polarization switching and saturation¹¹⁻¹⁴. In contrast, thin Zr-doped HfO₂ (HZO) exhibits outstanding ferroelectric performance at low driving voltages even at a thickness of merely 5nm²⁷⁻²⁹, which is crucial for scalable integrated circuits incorporating ultra-thin oxide semiconductors.”

24 Shen, C.-K., Chaurasiya, R., Chen, K.-T. & Chen, J.-S. Synaptic Emulation via Ferroelectric P (VDF-TrFE) Reinforced Charge Trapping/Detrapping in Zinc–Tin Oxide Transistor. *ACS Applied Materials & Interfaces* **14**, 16939-16948 (2022).

25 Lee, Y. T. *et al.* Nonvolatile ferroelectric memory circuit using black phosphorus nanosheet-based field-effect transistors with P (VDF-TrFE) polymer. *Acs Nano* **9**, 10394-10401 (2015).

26 Chu, F. J., Chen, Y. C., Shih, L. C., Mao, S. C. & Chen, J. S. Reconfigurable Physical Reservoir Enabled by Polarization of Ferroelectric Polymer P (VDF–TrFE) and Interface Charge-Trapping/Detrapping in Dual-Gate IGZO Transistor. *Advanced Functional Materials*, 2310951 (2023).

27 Ali, T. *et al.* in *65th IEEE Annual International Electron Devices Meeting (IEDM)*. (2019).

28 Kim, S. J. *et al.* Low-voltage operation and high endurance of 5-nm ferroelectric Hf_{0.5}Zr_{0.5}O₂ capacitors. *Applied Physics Letters* **113** (2018).

29 Lyu, X. *Polarization and Switching Dynamics Study of Ferroelectric Hafnium Zirconium Oxide for Feram and Fefet Applications*. (2023).

Comment 2: “And they also suffer from lower durability, typically around 10³ cycles.” This statement is not correct. In flexible FeFETs, even 10⁸ cycles of operation has been demonstrated. For instance, please check the state-of-the-art numbers from several different flexible FeFET structures from this work: <https://onlinelibrary.wiley.com/doi/full/10.1002/aisy.202100175>.

Response 2: Thank you for your valuable comment. We have reviewed the literature you provided, as well as the latest reports on flexible FeFETs, and have corrected the endurance description in the introduction to make the expression of our article more rigorous. We have cited the paper you provided in our revised manuscript and added the latest reports on flexible FeFETs. The revised parts and added discussion are as follows:

"Currently, whether it is two-terminal devices like FRAM or three-terminal transistors integrated with IGZO channels, the annealing temperature of HZO has been shown to be compatible with BEOL conditions^{19,30-32}. In addition, FeFETs based on HZO have demonstrated synaptic characteristics³³. On the other hand, the ferroelectric properties of HZO films fabricated on flexible mica substrates have continued to exhibit exceptional performance³⁴. Nonetheless, achieving high-performance HZO-based ferroelectric memories on flexible substrates remains challenging."

19 Sun, C. *et al.* Temperature-dependent operation of InGaZnO ferroelectric thin-film transistors with a metal-ferroelectric-metal-insulator-semiconductor structure. *IEEE Electron Device Letters* **42**, 1786-1789 (2021).

30 Luo, Q. *et al.* A highly CMOS compatible hafnia-based ferroelectric diode. *Nature Communications* **11** (2020). <https://doi.org/10.1038/s41467-020-15159-2>

31 Du, Y. *et al.* in *2023 IEEE Symposium on VLSI Technology and Circuits (VLSI Technology and Circuits)*. 1-2 (IEEE).

32 Zheng, Z. *et al.* BEOL-Compatible MFMIS Ferroelectric/Anti-Ferroelectric FETs-Part I: Experimental Results With Boosted Memory Window. *Ieee Transactions on Electron Devices* (2023).

33 Sun, C. *et al.* in *2022 International Electron Devices Meeting (IEDM)*. 2.1. 1-2.1. 4 (IEEE).

34 Xiao, W. *et al.* Thermally stable and radiation hard ferroelectric Hf_{0.5}Zr_{0.5}O₂ thin films on muscovite mica for flexible nonvolatile memory applications. *ACS Applied Electronic Materials* **1**, 919-927 (2019).

The revised parts are highlighted in the manuscript.

Corresponding change in manuscript: Yes

Location of Change:

- On Page 4,

“Currently, whether it is two-terminal devices like FRAM or three-terminal transistors integrated with IGZO channels, the annealing temperature of HZO has been shown to be compatible with BEOL conditions^{19,30-32}. In addition, FeFETs based on HZO have demonstrated synaptic characteristics³³. On the other hand, the ferroelectric properties of HZO films fabricated on flexible mica substrates have continued to exhibit exceptional performance³⁴. Nonetheless, achieving high-performance HZO-based ferroelectric memories on flexible substrates remains challenging.”

19 Sun, C. *et al.* Temperature-dependent operation of InGaZnO ferroelectric thin-film transistors with a metal-ferroelectric-metal-insulator-semiconductor structure. *IEEE Electron Device Letters* **42**, 1786-1789 (2021).

30 Luo, Q. *et al.* A highly CMOS compatible hafnia-based ferroelectric diode. *Nature Communications* **11** (2020). <https://doi.org/10.1038/s41467-020-15159-2>

31 Du, Y. *et al.* in *2023 IEEE Symposium on VLSI Technology and Circuits (VLSI Technology and Circuits)*. 1-2 (IEEE).

32 Zheng, Z. *et al.* BEOL-Compatible MFMIS Ferroelectric/Anti-Ferroelectric FETs-Part I: Experimental Results With Boosted Memory Window. *Ieee Transactions on Electron Devices* (2023).

33 Sun, C. *et al.* in *2022 International Electron Devices Meeting (IEDM)*. 2.1. 1-2.1. 4 (IEEE).

34 Xiao, W. *et al.* Thermally stable and radiation hard ferroelectric Hf_{0.5}Zr_{0.5}O₂ thin films on muscovite mica for flexible nonvolatile memory applications. *ACS Applied Electronic Materials* **1**, 919-927 (2019).

Comment 3: With respect to Fig. 1, it is written that “Oxygen vacancies account for 14.96% and 17.84% of all oxygen-related chemical bonds, respectively.” How is this calculated? Can the authors explain the process in the SI? Is the oxygen vacancy contribution dependent on the stack? Can the oxygen vacancy concentration be

modified controllably with fab process parameters or stack designs? What would be the effect on performance both for ITO and HZO?

Response 3: Thank you for your comment. Fig. 1f and 1g show the XPS spectra for O 1s region of the ITO and HZO respectively. And Fig. S2 illustrates the complete XPS spectra of ITO film and HZO film. We used a software called CasaXPS specifically designed to calculate XPS oxygen vacancies account.

Fig.1 The XPS spectra for O 1s region of the (f) HZO film and (g) ITO channel. Oxygen vacancies account for 14.96% and 17.84% of all oxygen-related chemical bonds, respectively.

Supplementary Fig.2 XPS spectra of ITO film and HZO film. (a) XPS spectrum of ITO film, including counts of C1s, In 3d, Sn3d and O 1s. (b) XPS spectrum of HZO film, including counts of Hf 3f, Zr 3d, C 1s and O 1s.

First, the data is obtained by XPS measurement and identify the chemical state of O. Second, the data about O was imported into CasaXPS. The account ratio of different

binding energy of O (Metal-O, O-H/C-O and oxygen vacancy V_O) is obtained by calculating in the software with the help of help manual of CasaXPS⁶. Last, the data was exported and demonstrated in Fig. 1f and 1g.

The specific information about CasaXPS is added at SI.

The stacking of ITO and HZO will affect the oxygen vacancy concentration at the interface, but not on the surface of ITO. The concentration of oxygen vacancy in ITO can be changed by oxygen pressure during the PVD growth of ITO. The higher the oxygen pressure during PVD growth, the smaller the oxygen vacancy concentration of ITO¹⁷. The less O-H/C-O and more M-O will contribute to the reduced surface scattering^{7-9, 17}. At the same time, the oxygen vacancy concentration in ITO affects the V_{th} , SS, mobility and DIBL as so on of FeFETs¹⁷.

For ferroelectric materials, the concentration of oxygen vacancy has a great influence on the Pr and too large and too small oxygen vacancy concentration will result in the degradation of Pr¹⁰⁻¹². Too much oxygen vacancy will destroy the withstand voltage and lattice structure of HZO and less oxygen vacancy will make HZO have smaller Pr and larger E_c which is also unfavorable to the ferroelectric properties of HZO¹³. The concentration of oxygen vacancy in HZO can be changed through the growth of ALD, Plasma power and time. The second is to regulate the oxygen vacancy in the film during the annealing process. Annealing in oxygen atmosphere will reduce the oxygen vacancy concentration¹⁴. Therefore, appropriate oxygen vacancy concentration is crucial for ITO and HZO.

6 http://www.casaxps.com/help_manual/ from CasaXPS, 2013.

7 Yeob Park, S. et al. Improvement in the device performance of tin-doped indium oxide transistor by oxygen high pressure annealing at 150 °C. *Appl. Phys. Lett.* 100, 162108 (2012).

8 Deng, S. et al. Investigation of high-performance ITO-stabilized ZnO TFTs with hybrid-phase microstructural channels. *IEEE Trans. Electron Dev.* 64, 3174 - 3182 (2017).

9 Lee, J. et al. High mobility ultra-thin crystalline indium oxide thin film transistor using atomic layer deposition. *Appl. Phys. Lett.* 113, 112102 (2018).

10 Chen J, et al. Controlling the Ferroelectricity of Doped-HfO₂ via Reversible Migration of Oxygen Vacancy. *IEEE Transactions on Electron Devices* 70, 1789-1794 (2023).

11 Chen J, et al. Impact of Oxygen Vacancy on Ferroelectric Characteristics and Its Implication for Wake-Up and Fatigue of HfO₂-Based Thin Films. *IEEE Transactions on Electron Devices* 69, 5297-5301 (2022).

12 Islamov DR, Perevalov TV. Effect of oxygen vacancies on the ferroelectric Hf_{0.5}Zr_{0.5}O₂ stabilization: DFT simulation. *Microelectronic Engineering* 216, 111041 (2019).

13 Li Z, et al. The Doping Effect on the Intrinsic Ferroelectricity in Hafnium Oxide-Based Nano-Ferroelectric Devices. *Nano Letters* 23, 4675-4682 (2023).

14 Li Z, et al. Understanding the Effect of Oxygen Content on Ferroelectric Properties of Al-Doped HfO Thin Films. *IEEE Electron Device Letters* 44, 56-59 (2023).

17 Li, S. et al. Nanometre-thin indium tin oxide for advanced high-performance electronics. *Nature materials* 18, 1091-1097 (2019).

The revised parts are highlighted in the manuscript.

Corresponding change in manuscript: Yes

Location of Change:

- On Page 6,

“The calculation method for oxygen vacancies is presented in Supplementary Fig. 3. The concentration of oxygen vacancies can be controllably altered by adjusting process parameters^{23,37}. An appropriate concentration of oxygen vacancies is crucial for both ITO and HZO.”

23 Li, S. *et al.* Nanometre-thin indium tin oxide for advanced high-performance electronics. *Nature Materials* **18**, 1091-+ (2019).

37 Li, Z. *et al.* Understanding the Effect of Oxygen Content on Ferroelectric Properties of Al-Doped HfO Thin Films. *IEEE Electron Device Letters* **44**, 56-59 (2022).

- On page S4 of Supplementary Information,

Supplementary Fig. 3 The CasaXPS software usage diagram¹.

The calculation process is as follows.

1. Data importing. The XPS spectrum data is imported into the CasaXPS.
2. Peak recognition and distribution. CasaXPS analyzes the spectrum of O 1s region, and identifies different peaks according to different chemical states of oxygen, including the oxygen in metal oxides (M-O), the oxygen in hydroxyl groups or the oxygen in carbon oxygen groups (O-H/C-O) and oxygen vacancy (V_O). The binding energy of oxygen at each state is different, forming different peaks.

3. Peak fitting. CasaXPS uses the Gauss-Lorentz mixture function (or other fitting function) to fit the three peaks above. By adjusting the fitting parameters (such as peak position, width, shape, etc.), the fitting curve is made as close as possible to the experimental data.

4. Quantitative analysis and calculate the ratio: After fitting is completed, CasaXPS calculates account in various chemical states according to the area of the peak. The account of oxygen in the chemical state is directly proportional. The ratio of M-O, O-H/C-O and V_O is obtained.

1 http://www.casaxps.com/help_manual/ from CasaXPS, 2013.

Comment 4: What are the leakage current density of the devices? The leakage current of the HZO gate stack and the transistor gate current should be plotted.

Response 4: Thank you for your comment. The leakage current density of devices is crucial for the assessment of device reliability. We first measured the leakage current of a 12 nm thick HZO gate stack, as shown in Figure R1, with an inset schematic of the voltage applied to the gate stack. It can be clearly observed that a significant charging current is generated in the initial phase, followed by a stable leakage current level. The leakage current of the HZO gate stack remains below 1 nA, even under a 3V bias.

Moreover, the gate current (I_G) of the transistor more accurately reflects the state of the device during operation. As shown in Figure R2, the gate leakage current of the device remains below pA over a gate sweep range of -2V to 2V.

Figure R1 Leakage current of the HZO gate stack under different applied voltages.

Figure R2 Measured I_G - V_G curve, showing sub-pA. gate/substrate leakage.

The low leakage current levels of both the HZO gate stack and transistor gate ensure the reliability and stability of the device. We have added this information to the revised manuscript and supplementary information and highlighted it.

Corresponding change in manuscript: Yes

Location of Change:

- On Page 7,

“Magnified images of the transistor gate current and the leakage current of the HZO gate stack at different applied voltages are shown in Supplementary Fig. 6. The low level of leakage current ensures the reliability and stability of the device.”

- On page S6 of Supplementary Information,

Supplementary Fig.6 The leakage current of the HZO gate stack and the transistor gate current. (a) Measured I_G - V_G curve, showing sub-pA. gate/substrate leakage. (b) Leakage current of the HZO gate stack under different applied voltages.

Comment 5: A wafer-scale performance variation mapping would be important. At least a distribution of KPIs for more than 25-30 devices can provide an accurate picture. Prediction based on single device performance, is often over-optimistic.

Response 5: Thank you for your valuable comment. As you pointed out, wafer-scale performance variation mapping provides more effective and accurate information. We conducted a statistical analysis of the transfer characteristics curves within a $\pm 2V$ gate voltage (V_G) sweep range for 30 devices, resulting in a distribution graph of KPIs for these devices. This includes the on/off ratio (I_{ON}/I_{OFF}), subthreshold swing (S.S.), memory window (MW), and the on-state current (I_{ON}) among other critical performance metrics.

Figures R3a and R3b show the distribution of the on/off ratio and MW for the 30 devices, with 93.3% of the devices exhibiting an on/off ratio exceeding 10^7 , and 83.3% of the devices demonstrating a MW larger than 0.6 V. Additionally, statistical analysis of the S.S. and gate current (I_G) for these 30 devices revealed that over 80% of the devices had a S.S. smaller than 60 mV/dec. (Figure R4a), and the maximum I_G for all devices was only 3 pA, with 80% of the devices exhibiting I_G below the pA level (Figure R4b). Finally, analysis of the output characteristic curves of some devices showed that over 80% of the devices displayed a high I_{ON} exceeding $60\mu A/\mu m$ (Figure R5).

Figure R3 Distribution of (a) on/off ratio and (b) memory window for the devices.

Figure R4 Statistics of (a) subthreshold swing (S.S.) and (b) gate leakage current (I_G) for 30 devices.

Figure R5 Statistics of on-state current (I_{ON}) for the devices.

The distribution graphs of these KPIs demonstrate that the overall performance of the devices is excellent, with over 80% of the devices showing outstanding performance in various metrics. This is attributed to the mature device fabrication process and stable experimental conditions. We have added this information to the supplementary information to better illustrate the comprehensive assessment of device performance.

Corresponding change in manuscript: Yes

Location of Change:

- On Page 8,

“The distribution of key performance indicators (KPIs) for 30 devices is presented in Supplementary Fig. 8, including critical performance information such as the on/off ratio, S.S., MW, and I_{ON} . The overall performance of the devices is favorable, which can be attributed to mature device fabrication processes and stable experimental conditions.”

- On page S7 of Supplementary Information,

Supplementary Fig.8 The distribution of key performance indicators (KPIs) for 30 devices. Distribution of (a) on/off ratio and (b) memory window for the devices. Statistics of (c) subthreshold swing (S.S.) and (d) gate leakage current (I_G). (e) Statistics of on-state current (I_{ON}).

Comment 6: HZO gate stack response and FeFET performance under different bending radii would be important to have. Due to piezoelectric nature of the ferroelectric thin films, additional charge generation due to bending (and hence deformation of the FE) is expected. It would be important to know how they affect device properties like V_c , MW, Ion/off, endurance.

Response 6: Thank you for your comment. Following your suggestion, we have tested the performance of the HZO gate stack and FeFETs under different bending radii, as shown in Figures R6~R8. The results demonstrate that we have not only analyzed the devices at various bending radii but also after different numbers of bending cycles, including their impact on device properties such as remanent polarization (P_r), memory window (MW), Ion/off ratio, and endurance.

Figures R6a and R6b show the influence of different bending radii (unbent, 7 mm, 5 mm) and bending cycles (initial, 100 cycles, 500 cycles) on the residual polarization

characteristics of the HZO gate stack. After bending to a radius of 5 mm or after 500 bending cycles, only a slight decay in P_r is observed relative to the initial stage. Figures R7a and R7b present the on/off ratio and MW extracted from the transfer characteristic curves (insets) under different bending radii. No significant change in the on/off ratio was observed under various bending radii, and while some decay in the MW was observed at a bending radius of 5 mm, it still maintained a high level ($>0.6V$), with no significant increase in gate leakage current detected. Figures R8a and R8b show the effects of bending radius and number of bending cycles on the endurance of the ferroelectric transistors. It is evident that a bending radius of 7 mm and 500 bending cycles did not significantly affect the endurance of the devices.

Figure R6 (a) P-V hysteresis loops measured under various bending radii, and (b) P-V hysteresis loops measured under various bending cycles with 7 mm bending radius.

Figure R7 (a) On/off current ratio and (b) memory window measured in FeFET as a function of the bending radius. These measurements were performed at $V_D = 0.1$ V.

Figure R8 Performance of the endurance property for the ITO FeFETs with different (a) bend radii and (b) bend cycles.

The results indicate that the bending process does not cause significant damage to the structure of the HZO gate stack or the performance of the FeFETs. This is because the internal stress endured by the films during bending is greatly related to the film thickness; thicker films generate substantial internal stress when bent^{11,12}. The remarkable flexibility of the reported devices can be attributed to the ultra-thin HZO ferroelectric film and the ultra-thin ITO channel, which minimizes the strain caused by bending.

devices. *Advanced Functional Materials* **27**, 1700461 (2017).

12 Park, S. I. *et al.* Theoretical and experimental studies of bending of inorganic electronic materials on plastic substrates. *Advanced Functional Materials* **18**, 2673-2684 (2008).

We have added this information to the revised manuscript and supplementary information and highlighted it to enrich the content of our work.

Corresponding change in manuscript: Yes

Location of Change:

- On Page 12,

“Additionally, we have investigated the impact of different bending radii and bending cycles on the response of the HZO gate stack and the I_{ON}/I_{OFF} ratio, endurance, and other performances of FeFETs, as shown in Supplementary Fig. 10 and Fig. 11. The device performance remains stable under different application scenarios.”

- On page S8 of Supplementary Information,

Supplementary Fig.10 The performance of the HZO gate stack and FeFETs under different bending radii. (a) P-V hysteresis loops measured under various bending radii. (b) On/off current ratio and (c) memory window measured in FeFET as a function of the bending radius. These measurements were performed at $V_D = 0.1$ V. (d) Performance of the endurance property for the ITO FeFETs with different bend radii.

Comment 7: The training protocol is not clear from the discussion. most likely, it is based on single device performance extrapolated to form a crossbar? What would be the configuration and applied pulse protocols for the training? How does the size of the network and task complexity affect the classification accuracy? How could cycle-to-cycle and device-to-device performance variation could affect the network performance?

Response 7: We are grateful for your insightful comments. The multi-level pulse modulation characteristics of ITO FeFETs can emulate biological synaptic plasticity, facilitating network classification and recognition capabilities. Essentially, applying synaptic devices in artificial neural networks (ANNs) involves simulating synaptic weights through the device's multi-level conductance modulation. Inter-device variations can lead to differences in synaptic performance. However, in our FeFETs fabrication process, the uniform growth of HZO dielectric thin films and ITO channel materials using stable equipment across a large area ensures consistency in the conductance modulation range and synaptic weight changes among different synaptic devices. Like most previous reports, in neural network simulations based on FeFET synaptic devices, we use the conductance values of some devices to replace the synaptic weights in the ANN to achieve classification and recognition functions, ensuring accuracy without sacrificing efficiency^{1,2}. Additionally, the development of high-performance network algorithms and the construction of reliable ANNs are equally crucial.

In neuromorphic computing, accuracy and efficiency are vital factors in assessing network quality. Figure R9a illustrates a three-layer ANN utilizing the backpropagation (BP) algorithm to recognize handwritten digit images of 16×16 pixels from the revised MNIST database, with some examples shown in Figure R9b.

Figure R9 (a) Schematic of the neural network used for recognizing handwritten digits with a resolution of 16×16 pixels, featuring a single hidden layer. (b) Examples of handwritten digits from the MNIST training dataset.

The ANN built in this study consists of an input layer, a hidden layer, and an output layer to recognize handwritten digit images. The input layer has 256 neurons corresponding to the 256 (16×16) pixels. The number of neurons in the hidden layer impacts network accuracy, which will be discussed later. The output layer has 10 neurons for digits 0~9. The weights between the input and hidden layers are denoted by V , and those between the hidden and output layers by W .

The training of the ANN includes two main steps: signal forward propagation and error backpropagation to achieve an optimized weight network. The ANN training and recognition flowchart for digit images is shown in Figure R10. N represents the number of training epochs, set to a maximum of 1000. n is the number of images trained per epoch, with a maximum of 10000. ΔW is the weight change after each epoch. The input to the input layer is X_n , the output of the output layer is O_K , with y_k as the ideal output. The training process begins by calculating the hidden layer's input I_j , followed by the

log-sigmoid non-linear activation function to obtain the hidden layer's output M_j , and finally, calculating the output layer's output O_k , comparing it with y_k . Feedback is given at the end of an epoch to obtain ΔW . After adjusting V and W , a new epoch begins until training concludes.

Figure R10 Flowchart illustrating the training process of the neural network for digit image recognition.

The size of the network and task complexity significantly impact ANN classification accuracy and efficiency, reflected in the number of hidden layer neurons and the initial settings of V and W . We investigated the relationship between network recognition accuracy and hidden layer neuron numbers as shown in Figure R11a. Increasing the number of neurons (n) in the hidden layer necessitates more computational resources and time. However, when n is too small, the network's accuracy is insufficient. We trained the network with three different n values, finding the highest final accuracy at $n=64$, which also ensured training efficiency.

The initial settings of V and W also influence the network's recognition accuracy and efficiency. We set V and W to either a normal or random distribution, varying the standard deviation ($\delta=0.1, 0.2, 1$) under normal distribution. As depicted in Figure R11b, when $\delta=0.2$, the network achieved the highest final recognition rate and required the fewest epochs to reach 90% accuracy. Therefore, in our studies of ANNs built on FeFETs, we set the initial values of V and W to a normal distribution with $\delta=0.2$.

Figure R11 (a) Impact of the number of neurons in the hidden layer on the network's recognition accuracy. (b) Effect of the initial values of V and W on recognition accuracy.

In practical applications, each synaptic device corresponds to a neuron in the ANN. Imperfections in fabrication processes introduce variability between cycles and devices, introducing noise into the network. The presence of noise significantly affects the ANN's accuracy and can render the network ineffective in severe cases. To validate the fault tolerance of our constructed ANN, we randomly introduced different proportions of noise pixels into the training set, examining the relationship between recognition accuracy and noise pixel ratio as shown in Figure R12.

Figure R12 Relationship between the network's recognition accuracy and the proportion of noise.

With increasing noise pixel ratios, the network's recognition accuracy gradually declined. When the noise pixel ratio reached 70%, accuracy fell below 85%. At this point, we believe the network loses its functionality for image recognition and classification. Our studies indicate that our optimizations of the network's initial state are successful, providing strong fault tolerance. This lays a solid foundation for more complex neural processing in future work.

1 Li, Q. *et al.* Organic optoelectronic synaptic devices for energy-efficient neuromorphic computing. *IEEE Electron Device Letters* **43**, 1089-1092 (2022).

2 Cao, Y. *et al.* An Efficient Training Methodology of Hardware Neural Network Based on Wafer-Scale MoS₂ Synaptic Array. *Advanced Electronic Materials* **8** (2022).

We hope this comprehensive response addresses your concerns and elucidates our network's capabilities. We have added this information to the revised manuscript and supplementary information and highlighted it to enrich the content of our work.

Corresponding change in manuscript: Yes

Location of Change:

- On Page 13,

“The process of training and recognizing digital images using ANN^{63,64}, including signal forward propagation and error backpropagation, is depicted in Supplementary Fig. 13. The relationship between network recognition accuracy and factors such as the number of neurons in the hidden layer and the initial values of weights V and W, which significantly influence the accuracy and efficiency of ANN classification, is shown in Supplementary Fig. 14. The correlation between recognition accuracy and the proportion of noise pixels is demonstrated in Supplementary Fig. 15, illustrating the robust fault tolerance of the constructed ANN”

63 Li, Q. *et al.* Organic optoelectronic synaptic devices for energy-efficient neuromorphic computing. *IEEE Electron Device Letters* **43**, 1089-1092 (2022).

64 Cao, Y. *et al.* An Efficient Training Methodology of Hardware Neural Network Based on Wafer-Scale MoS₂ Synaptic Array. *Advanced Electronic Materials* **8** (2022).

- On page S10 and page S11 of Supplementary Information,

Supplementary Fig.13 Flowchart illustrating the training process of the neural network for digit image recognition.

Supplementary Fig.14 The relationship between network recognition accuracy and ANN parameters. (a) Impact of the number of neurons in the hidden layer on the network's recognition accuracy. (b) Effect of the initial values of V and W on recognition accuracy.

Supplementary Fig.15 Relationship between the network's recognition accuracy and the proportion of noise.

Comment 8: Multiple synaptic plasticity with varied time constants is mentioned in the article. Can a network based on these FeFETs able to perform spatio-temporally varying dataset?

Response 8: Thank you for your comment. To handle spatio-temporally varying datasets, a neural network should be capable of processing information across different

time scales simultaneously. Networks based on ferroelectric field-effect transistors (FeFETs), due to their synaptic plasticity and high-pass filtering characteristics, can theoretically process such types of data.

The behaviors of FeFETs across various time scales can assist the network in capturing both short-term and long-term dynamics in the data. Different time constants allow the network to pay attention to both immediate events and long-term trends, which is critical for understanding and predicting complex spatiotemporal relationships. The high-pass filtering ability of the FeFET network can differentiate the rapidly changing parts of the input signal, such as movement in a video, which is a key capability for processing spatiotemporal data. Moreover, the deployment of multiple time constants enables the network to adjust its response over various time scales through learning, thereby enhancing its generalization capability to unseen spatiotemporal datasets.

Additionally, the synaptic plasticity of FeFETs has shown reliability under different bending radii or bending cycles, suggesting that networks based on FeFETs can maintain their performance despite physical changes. This is paramount for processing dynamic and changing datasets.

Therefore, networks based on FeFETs are capable of executing spatiotemporally varying datasets, benefiting from the remarkable synaptic plasticity of FeFETs that allows them to react to temporal changes while maintaining consistent performance across different physical states. In practical applications, identifying the optimal time constants for a specific task may require experimentation and tuning. For instance, in recurrent neural networks (RNNs) or long short-term memory networks (LSTMs), time constants could be adjusted via the network architecture (such as settings of the forget gates) and training parameters (like learning rates and weight initialization).

Thanks a lot for your suggestions.

Response to Reviewer 2's Comments:

Comment: The paper reports outstanding results of implemented FeFET on transparent and flexible substrates. We recommend major revision in order to complete the full picture and also clearing minor typos.

Response: We appreciate the reviewer's valuable time for evaluating our manuscript. We appreciate reviewer's positive evaluation of our study as well as important comments, which have helped to improve our manuscript. Detailed responses are provided in the following sections. We hope the revised manuscript could meet the quality require of Nature Communications. We thank the reviewer again for the very good suggestions.

Comment 1: The paper states excellent performance values of implemented FeFET. However, background information as well as micrograph about device geometry is missing and would be helpful to understand and compare the device performance.

Response 1: Thank you for your comment. The device geometry is a very important parameter. Fig. R13 illustrates the shape of gate and source/drain with ITO channel. And the channel length and width are 5 μ m and 50 μ m respectively.

Figure R13 The micrograph of ITO FeFET.

The revised parts are highlighted in the manuscript.

Corresponding change in manuscript: Yes

Location of Change:

- On page 6,

“The micrograph of the device is shown in Supplementary Fig. 1, displaying the shape of gate and source/drain with ITO channel. The channel length and width are 5 μm and 50 μm , respectively.”

- On page S4 of Supplementary Information,

Supplementary Fig.1 The micrograph of ITO FeFET.

Comment 2: Further it would be worth to discuss impact of pulse width on program / erase performance including discussion about potential impact of trapping on the switching performance. Based on the subloop switching behavior and matching with the PV characteristic no saturation in Fig 2f is visible which points to a high risk of trapping as prevailing switching mechanism rather than FE switching.

Response 2: Thank you for your insightful comment. The HfO_2 material is characterized by a high density of intrinsic defects (ranging from 10^{-12} to 10^{-14} cm^{-2}), including oxygen vacancies and interstitial oxygen atoms that can act as traps for electrons or holes^{1,2}. Understanding the potential impact of these traps on switching

performance is crucial for enhancing the stability of transistors.

The effect of charge trapping on the performance of HfO₂-based FeFETs is expected. As shown in Figure R14, charge trapping leads to a shift in the threshold voltage (V_{TH}), which is opposite to the V_{TH} shift caused by ferroelectric switching under the same gate voltage polarity. Thus, the combination of these two mechanisms results in a reduction of the FeFET device's memory window^{3,4}. In our ITO FeFETs, the transfer characteristics demonstrate a counterclockwise hysteresis, a signature of ferroelectric switching⁵.

Figure R14 (a) FeFET NVM operation. (b) Electron/hole trapping within gate-stack during erase/program pulse, respectively. (c) Charge trapping narrows MW. (d) V_{TH} distribution without (solid line) and with (dashed line) charge trapping. Charge trapping broadens and shifts distribution³.

Several studies have explored the influence of pulse width on the programming/erasing characteristics of FeFET devices. Ekaterina et al. utilized a single-pulse I_D - V_G technique to study the interplay between ferroelectric switching and charge trapping in Si: HfO₂-based FeFET devices⁶. The applied pulse sequence, as depicted in Figure R15a, begins with a -6V negative pulse to establish a saturated negative polarization state, followed by two positive pulses with a one-minute interval. The V_{TH} shifts observed between the rising (V_{TH1}) and falling (V_{TH2}) edges of the I_D - V_G characteristics from these pulses help to identify the dominant mechanism, either ferroelectric switching ($V_{th}<0$) or parasitic electron trapping ($V_{th}>0$). Figure R15b

shows the results of trap tests with single pulses at +4V amplitude and different pulse widths of 0.5, 1, and 10 μs . For all pulse widths, the I_D - V_G characteristics obtained on the falling edge (2) of the first pulse displayed a positive shift compared to those on the rising edge (1), indicating electron trapping as the predominant mechanism. However, after a delay of one minute, the rising edge of the second pulse (3) showed a negative shift compared to the initial characteristics (1), which can only be explained by ferroelectric polarization and not solely by electron detrapping, especially when detrapping occurs at zero gate voltage. Additionally, by altering the pulse amplitude (from 2 to 5V) and width (from 0.1 to 100 μs), the direct V_{TH} shift observed after the first (V_{TH12}) and second (V_{TH14}) positive pulses, and after a one-minute delay (V_{TH3}), was correlated with the single-pulse width (t_{TP}) in Figure R15c. When the amplitude exceeds 2V, V_{TH12} is positive, indicating that trapped electron charges are directly generated after applying the positive pulse, with larger pulse amplitude and width leading to a greater positive shift in V_{TH} , suggesting an increase in the number of captured electrons. On the other hand, V_{TH13} remains negative for all pulses, indicating that the ferroelectric polarization charge surpasses the captured electron charge, with detrapping commencing after a one-minute delay.

Figure R15. Study of the superposition of the ferroelectric switching and charge trapping in Si: HfO₂-based FeFET cells. (a) Experimental gate pulse sequence. (b) I_D - V_G characteristics measured on the rising (1) and falling (2) edges of the first single-

pulse as well as after a delay of 1 min (3) for positive pulse width of 0.5, 1 and 10 μ s. (c) V_{TH} shift directly after the pulse (V_{TH12}), after a delay of 1 min (V_{TH13}) and directly after the second positive pulse (V_{TH14}) as a function of pulse width for different pulse amplitudes⁶.

The saturation characteristic of ferroelectric polarization and the non-saturating behavior of charge trapping explain the non-linear behavior of the memory window in terms of pulse width and amplitude, consistent with previous studies on HfO₂-based FeFETs^{2,4,7}. In our ITO FeFETs, the combination of charge trapping and ferroelectric switching is considered a plausible explanation for the observed memory window behavior, which can be further substantiated through different measurement techniques such as transient or charge pumping experiments⁸.

The charge trapping and generation of interface traps (including border traps) significantly impact key memory characteristics like data retention and endurance^{1,7,9}. To fully mitigate parasitic trapping during device operation, modifications to the gate stack structure or the use of HfO₂ layers with lower ferroelectric polarization are necessary^{4,6}. Additionally, trapping has been confirmed as a transient phenomenon, with more trapping occurring in the oxide over longer measurement times². Thus, variations in measurement speed result in changes in the net trapped charge, consequently affecting the trapping component of V_{TH} . Trapping and depolarization produce V_{TH} shifts of opposite signs, and at appropriate measurement speeds, these two components may compensate each other, resulting in a noticeable zero hysteresis. However, in reality, as measurement speed increases, Lee et al. observed greater counterclockwise hysteresis (less trapping/more depolarization)¹⁰.

Figure R16. P-V loops of representative capacitors with (a) W/11.4 nm HZO /Ni and (b) W/11.4 nm HZO/3.4 nm ITO/Ni. The similar P-V characteristics suggest that the interface of FE and channel is excellent. (c) Retention characterization of the ITO FeFETs. Optimized retention extrapolated as >10 yrs. at $V_D = 0.1$ V. (d) Performance of the endurance property for the ITO FeFETs. High endurance exceeding 2×10^7 cycles.

The reported ITO FeFET exhibits excellent data retention and endurance characteristics (Figure R16c and R16d), attributed to the good interface between the ultra-thin ITO channel and HZO gate stack, as also evidenced in the P-V loops in Figure R16a and R16b. In this study, the memory window is affected by the combined action of charge trapping and ferroelectric switching, with measurement speed influencing the net trapped charge and thus the V_{TH} shift. Additionally, charge trapping reduces the effective voltage across the FE, forcing the FeFET to operate on a non-saturating polarization loop, hence no saturation is observed in Figure 2f. However, the counterclockwise I_D - V_G loop indicates that ferroelectric switching predominates.

- 1 Zeng, B. *et al.* Program/erase cycling degradation mechanism of HfO₂-based FeFET memory devices. *IEEE Electron Device Letters* **40**, 710-713 (2019).
- 2 Alam, M. N. K. *et al.* On the characterization and separation of trapping and ferroelectric behavior in HfZrO FET. *IEEE Journal of the Electron Devices Society* **7**, 855-862 (2019).
- 3 Ni, K. *et al.* Critical role of interlayer in Hf_{0.5}Zr_{0.5}O₂ ferroelectric FET nonvolatile memory performance. *IEEE Transactions on Electron Devices* **65**, 2461-2469 (2018).
- 4 Deng, S. *et al.* Unraveling the dynamics of charge trapping and de-trapping in ferroelectric FETs. *IEEE Transactions on Electron Devices* **69**, 1503-1511 (2022).
- 5 Ali, T. *et al.* High endurance ferroelectric hafnium oxide-based FeFET memory without retention penalty. *IEEE Transactions on Electron Devices* **65**, 3769-3774 (2018).
- 6 Yurchuk, E. *et al.* Charge-trapping phenomena in HfO₂-based FeFET-type nonvolatile memories. *IEEE Transactions on Electron Devices* **63**, 3501-3507 (2016).
- 7 Yurchuk, E. *et al.* in *2014 IEEE International Reliability Physics Symposium*. 2E. 5.1-2E. 5.5 (IEEE).
- 8 Mulaosmanovic, H., Breyer, E. T., Mikolajick, T. & Slesazeck, S. Ferroelectric FETs with 20-nm-thick HfO₂ layer for large memory window and high performance. *IEEE Transactions on Electron Devices* **66**, 3828-3833 (2019).
- 9 Gong, N. & Ma, T.-P. A study of endurance issues in HfO₂-based ferroelectric field effect transistors: Charge trapping and trap generation. *IEEE Electron Device Letters* **39**, 15-18 (2017).
- 10 Lee, M. *et al.* in *2017 IEEE International Electron Devices Meeting (IEDM)*. 23.23. 21-23.23. 24 (IEEE).

Comment 3: Also I think there is a mismatch between Fig 2f and 2h.

Response 3: Thank you for your comment. Fig. 2f and 2h show I_D-V_G curves for ITO FeFETs and threshold voltages after erase and program versus V_G Sweep range of ITO FeFETs at V_D=0.1 V respectively. The V_{th} shown in Fig. 2h is presented by tangential

method in linear scale. In order to match Fig. 2f, we change Fig. 2f to the on-state current method in log scale to extract V_{th} (Figure R17). V_{th} is extracted at I_D of 10^{-8} A/ μ m.

Figure R17. Threshold voltages after erase and program versus V_G Sweep range of ITO FeFETs at $V_D=0.1$ V.

The revised parts are highlighted in the manuscript.

Corresponding change in manuscript: Yes

Location of Change:

- On page 23,

Fig.2 Hysteresis and electrical properties of ITO FeFETs. P-V loops of representative capacitors with (a) W/11.4 nm HZO /Ni and (b) W/11.4 nm HZO/3.4 nm ITO/Ni. The similar P-V characteristics suggest that the interface of FE and channel is excellent. (c) Measured I_D - V_G of ITO FeFETs exhibiting near-ideal S.S. value, high I_{ON}/I_{OFF} ratio of 10^8 , and sub-pA. gate/substrate leakage. (d) Measured I_D - V_D curve, showing high I_{ON} of $105 \mu\text{A}/\mu\text{m}$ at $V_G = 3 \text{ V}$ & $V_D = 0.1 \text{ V}$. (e) The S.S. distribution of ITO FeFETs devices, showing minimum S.S. of $33 \text{ mV}/\text{decade}$. (f) I_D - V_G curves for ITO FeFETs. A stable FE type hysteresis with MW up to 2.78 V is available. (g) Memory window (MW) versus $V_{G, \text{MAX}}$ of ITO FeFETs at $V_D = 0.1 \text{ V}$. MW is calculated as ΔV_{th} in Fig. 2f. Record-high MW of 2.78 V at $V_{G, \text{MAX}} = 5 \text{ V}$. (h) Threshold voltages after erase and program versus V_G Sweep range of ITO FeFETs at $V_D = 0.1 \text{ V}$. (i) Benchmarking of MW and I_{ON}/I_{OFF} performance of FeFETs reported in this work (shown as a five-pointed star) versus recently reported FeFETs (shown as other symbols), where the MW is normalized with respect to the ferroelectric layer thickness.

Comment 4: In the neuromorphic section I recommend a better differentiation between experimental and simulated results.

Response 4: Thank you for your suggestion. Enhancing the differentiation between experimental and simulated results in the neuromorphic computing section will indeed improve the coherence and readability of our manuscript.

In the revised version, we have added a new heading, "Artificial neural networks based on ITO FeFET synaptic devices," to clearly delineate the simulated results pertaining to neuromorphic computing. Additionally, we have modified the original title of the experimental section from "Characterization of synaptic properties of flexible ITO FeFETs " to "Experimental characterization of synaptic properties of flexible ITO FeFETs" to better specify the experimental nature of these findings.

The revised parts are highlighted in the manuscript.

Corresponding change in manuscript: Yes

Location of Change:

- On page 10,

“Experimental characterization of synaptic properties of flexible ITO FeFETs

The neuromorphic computing properties of the ITO FeFETs under different bending test conditions are studied. Fig. 4a is a schematic diagram of the test of a flexible device under a certain bending radius.....”

- On page 13,

“Artificial neural networks based on ITO FeFET synaptic devices

We constructed an artificial neural network (ANN) using ITO FeFETs to recognize handwritten digits in the MNIST database (Supplementary Fig. 6). The network consists of an input layer (784 neurons), a hidden layer (64 neurons), and an output layer (10 neurons)⁴⁴.....”

Thanks a lot for your suggestions.

Response to Reviewer 3's Comments:

Comment 1: In the work "High-performance ferroelectric field-effect transistors with ultra-thin indium tin oxide channels for flexible and transparent electronics" the authors demonstrate for the first time the integration of an HZO based oxide-channel FET on a flexible substrate of mica with an annealing at 400°C.

Even though, this is an interesting further progress with novel results on the FET properties, particularly also its synaptic properties, there has been extensive work already in the field demonstrating:

- HZO annealing for BEOL compatible integration with HZO annealing at or below 400°C with a multitude of FRAM as well as FeMFET demonstrations
- the integration of HZO on IGZO channel with such a temperature profile compatible to BEOL conditions has been demonstrated on Si substrates
- the bending resistance of HZO based RRAM cells has been demonstrated on mica substrates before
- synaptic behavior in similar ferroelectric FETs

Hence, some clarifications could be helpful to highlight the key improvement of the manuscript.

Response 1: Thank you for your constructive and encouraging comments regarding our manuscript. Following your suggestion, we have added clarifications and updates on the latest research developments in the revised version of our manuscript. The revised parts have been highlighted in the manuscript. We have also included pertinent references to underscore the importance and the main challenges associated with our work, which helps to strengthen the introduction. The additional references and discussions are as follows:

"Currently, whether it is two-terminal devices like FRAM or three-terminal transistors integrated with IGZO channels, the annealing temperature of HZO has been shown to be compatible with BEOL conditions^{19,30-32}. In addition, FeFETs based on HZO have demonstrated synaptic characteristics³³. On the other hand, the ferroelectric properties of HZO films fabricated on flexible mica substrates have continued to exhibit

exceptional performance³⁴. Nonetheless, achieving high-performance HZO-based ferroelectric memories on flexible substrates remains challenging."

19 Sun, C. *et al.* Temperature-dependent operation of InGaZnO ferroelectric thin-film transistors with a metal-ferroelectric-metal-insulator-semiconductor structure. *IEEE Electron Device Letters* **42**, 1786-1789 (2021).

30 Luo, Q. *et al.* A highly CMOS compatible hafnia-based ferroelectric diode. *Nature Communications* **11** (2020).

31 Du, Y. *et al.* in *2023 IEEE Symposium on VLSI Technology and Circuits (VLSI Technology and Circuits)*. 1-2 (IEEE).

32 Zheng, Z. *et al.* BEOL-Compatible MFMIS Ferroelectric/Anti-Ferroelectric FETs-Part I: Experimental Results With Boosted Memory Window. *Ieee Transactions on Electron Devices* (2023).

33 Sun, C. *et al.* in *2022 International Electron Devices Meeting (IEDM)*. 2.1. 1-2.1. 4 (IEEE).

34 Xiao, W. *et al.* Thermally stable and radiation hard ferroelectric Hf_{0.5}Zr_{0.5}O₂ thin films on muscovite mica for flexible nonvolatile memory applications. *ACS Applied Electronic Materials* **1**, 919-927 (2019).

The revised parts are highlighted in the manuscript

Corresponding change in manuscript: Yes

Location of Change:

- On page 4,

“Currently, whether it is two-terminal devices like FRAM or three-terminal transistors integrated with IGZO channels, the annealing temperature of HZO has been shown to be compatible with BEOL conditions^{19,30-32}. In addition, FeFETs based on HZO have demonstrated synaptic characteristics³³. On the other hand, the ferroelectric properties of HZO films fabricated on flexible mica substrates have continued to exhibit exceptional performance³⁴. Nonetheless, achieving high-performance HZO-based ferroelectric memories on flexible substrates remains challenging.”

19 Sun, C. *et al.* Temperature-dependent operation of InGaZnO ferroelectric thin-film transistors with a metal-ferroelectric-metal-insulator-semiconductor structure. *IEEE Electron Device Letters* **42**, 1786-1789 (2021).

30 Luo, Q. *et al.* A highly CMOS compatible hafnia-based ferroelectric diode. *Nature Communications* **11** (2020).

31 Du, Y. *et al.* in *2023 IEEE Symposium on VLSI Technology and Circuits (VLSI Technology and Circuits)*. 1-2 (IEEE).

32 Zheng, Z. *et al.* BEOL-Compatible MFMIS Ferroelectric/Anti-Ferroelectric FETs-Part I: Experimental Results With Boosted Memory Window. *Ieee Transactions on Electron Devices* (2023).

33 Sun, C. *et al.* in *2022 International Electron Devices Meeting (IEDM)*. 2.1. 1-2.1. 4 (IEEE).

34 Xiao, W. *et al.* Thermally stable and radiation hard ferroelectric Hf_{0.5}Zr_{0.5}O₂ thin films on muscovite mica for flexible nonvolatile memory applications. *ACS Applied Electronic Materials* **1**, 919-927 (2019).

Comment 2: Furthermore, I would be interested what is the expectation of the authors on the influence of the in-plane stress on the ferroelectric properties of the material? There have been reports, that the in-plane stress has a large influence in the formation of ferroelectric/antiferroelectric properties upon crystallization.

Response 2:

Thank you for your comment. For the in-plane stress (RTA condition and different electrodes), we expect that HZO can reach a large remanent polarization after annealing in N₂ atmosphere.

In terms of the effect of RTA temperature, low-temperature annealing leads to the low crystallinity of ferroelectric O-phase and then the ferroelectric performance degradation of HZO²². RTA time also can affect the remanent polarization²³. The annealing temperature down to 350 °C~400 °C (compatible with BEOL) and 30 s is sufficient to crystallize ferroelectric film²⁴⁻²⁵. And lower temperature annealing, such as 400°C, can effectively suppress the formation of non-ferroelectric phases and reduce leakage

current, leading to preferable ferroelectric properties²⁹. High temperature annealing will lead to serious HZO leakage³⁰.

As for the electrode, the top electrode mainly induces mechanical stress and the interface between HZO thus affecting P_r ²⁶. Different electrodes have different expansions coefficients which will affect P_r ²⁷. TiN is a common electrode in ferroelectric materials^{27, 28}. Compared with other common electrodes, TiN can make ferroelectric thin films have large P_r and low leakage current²⁷. Considering the growth of semiconductor ITO after removal of top electrode, TiN wet-etched easily was selected. Therefore, the ferroelectric film growth method introduced in this paper is used and the P-V loops were obtained in Figure 2a and 2b.

22. Wang Y, et al. Precrystallization Engineering of $Hf_{0.5}Zr_{0.5}O_2$ Film in Back-End-of-Line Compatible Ferroelectric Device for Enhanced Remnant Polarization and Endurance. *IEEE Electron Device Letters* 44, 396-399 (2023).

23. Lehninger D, et al. Back - End - of - Line Compatible Low - Temperature Furnace Anneal for Ferroelectric Hafnium Zirconium Oxide Formation. *Physical Status Solidi (A)* 217, 1900840 (2020).

24. Shibayama S, Nishimura T, Migita S, Toriumi A. Thermodynamic control of ferroelectric-phase formation in $Hf_xZr_{1-x}O_2$ and ZrO_2 . *Journal of Applied Physics* 124, 184101 (2018).

25. Hur J, Luo Y-C, Tasneem N, Khan AI, Yu S. Ferroelectric Hafnium Zirconium Oxide Compatible with Back-End-of-Line Process. *IEEE Transactions on Electron Devices* 68, 3176-3180 (2021).

26. Goh Y, Cho SH, Park S-HK, Jeon S. Crystalline Phase-Controlled High-Quality Hafnia Ferroelectric with RuO_2 Electrode. *IEEE Transactions on Electron Devices* 67, 3431-3434 (2020).

27 Cao R, et al. Effects of Capping Electrode on Ferroelectric Properties of $Hf_{0.5}Zr_{0.5}O_2$ Thin Films. *IEEE Electron Device Letters* 39, 1207-1210 (2018).

28 Li Z, et al. Stabilizing the Ferroelectric Phase in HfAlO Ferroelectric Tunnel Junction with Different Bottom Electrodes. *IEEE Electron Device Letters* 44, 947-950 (2023).

29 Kim SJ, et al. Large Ferroelectric Polarization of TiN/Hf_{0.5}Zr_{0.5}O₂/TiN Capacitors due to Stress-induced Crystallization at Low Thermal Budget. 111, 242901 (2017).

30 Jeon S, Das D, Gaddam V. Effect of High-Pressure annealing temperature on the ferroelectric properties of TiN/Hf_{0.25}Zr_{0.75}O₂/TiN capacitors. In: 2020 4th IEEE Electron Devices Technology & Manufacturing Conference (EDTM)), 2020.

The revised parts are highlighted in the manuscript and supporting information.

Corresponding change in manuscript: Yes

Location of Change:

- On Page 7,

“Different annealing temperatures and electrodes have a significant impact on the ferroelectric properties of HZO. Low-temperature annealing leads to a decline in ferroelectric performance³⁸, while excessively high annealing temperatures result in larger leakage currents³⁹. Since TiN is wet-etched easily and can endow the ferroelectric thin films with larger remnant polarization (Pr) and lower leakage current⁴⁰, TiN was chosen as the electrode for HZO annealing.”

38 Shibayama, S., Nishimura, T., Migita, S. & Toriumi, A. Thermodynamic control of ferroelectric-phase formation in Hf_xZr_{1-x}O₂ and ZrO₂. *Journal of Applied Physics* **124** (2018).

39 Jeon, S., Das, D. & Gaddam, V. in *2020 4th IEEE Electron Devices Technology & Manufacturing Conference (EDTM)*. 1-3 (IEEE).

40 Cao, R. *et al.* Effects of capping electrode on ferroelectric properties of Hf_{0.5}Zr_{0.5}O₂ thin films. *IEEE Electron Device Letters* **39**, 1207-1210 (2018).

Comment 3: More measurement data should be shown concerning the basic ferroelectric FET properties upon bending cycles. So far only the synaptic properties are shown.

Response 3: Thank you for your valuable comment. In response to your suggestion, we have conducted tests on the performance of the HZO gate stack and FeFETs across different bending cycles, and the results are presented in Figures R18~R20. The analysis covered not only devices under various bending cycles but also those under different bending radii, including their impact on residual polarization (P_r), on-state current (I_{on}), on/off ratio, and endurance.

Figures R18a and R18b illustrate the effects of different bending cycles (initial, 100 cycles, 500 cycles) and bending radii (unbent, 7 mm, 5 mm) on the polarization of the HZO gate stack. Even after bending to a radius of 5 mm or undergoing 500 bending cycles, the devices exhibit only a minor decay in P_r relative to the initial stage. Figure R19a shows the output characteristic curves of the FeFETs during different bending cycles, where the I_{on} remains high, approximately 500 μA . Figure R19b depicts the relationship between the on/off current ratio and the number of bending cycles in FeFETs, indicating no degradation in the on/off ratio even after 1000 bending cycles. Figures R20a and R20b respectively show the effects of the number of bending cycles and bending radii on the endurance of the FeFETs. It is evident that a bending radius of 7 mm and up to 500 bending cycles do not significantly affect the device endurance.

Figure R18 (a) P-V hysteresis loops measured under various bending cycles, and (b) P-V hysteresis loops measured under various bending radii.

Figure R19 (a) I_D - V_D curve under different gate voltage recorded after various bending cycles. (b) On/off current measured in FeFET as a function of the number of bending cycles. These measurements were performed at $V_D = 0.1$ V with a bending radius of 7 mm.

Figure R20 Performance of the endurance property for the ITO FeFETs with different (a) bend cycles and (b) bend radii.

The results suggest that the bending process does not cause significant damage to the structure of the HZO gate stack or the performance of the FeFETs. The internal stress endured by the films during bending is highly related to the film thickness; thicker films can generate substantial internal stress when bent^{11,12}. The remarkable flexibility of the reported devices can be attributed to the extremely thin HZO ferroelectric film and the

ultra-thin ITO channel, which minimizes the strain induced by bending.

11 Yu, H. *et al.* Flexible inorganic ferroelectric thin films for nonvolatile memory devices. *Advanced Functional Materials* **27**, 1700461 (2017).

12 Park, S. I. *et al.* Theoretical and experimental studies of bending of inorganic electronic materials on plastic substrates. *Advanced Functional Materials* **18**, 2673-2684 (2008).

We have incorporated this information into the revised manuscript and supplementary information and highlighted it to enrich the research content.

Corresponding change in manuscript: Yes

Location of Change:

- On Page 12,

“Additionally, we have investigated the impact of different bending radii and bending cycles on the response of the HZO gate stack and the I_{ON}/I_{OFF} ratio, endurance, and other performances of FeFETs, as shown in Supplementary Fig. 10 and Fig. 11. The device performance remains stable under different application scenarios.”

- On page S9 of Supplementary Information,

Supplementary Fig.11 The performance of the HZO gate stack and FeFETs under different bending cycles. (a) P-V hysteresis loops measured under various bending cycles. (b) I_D - V_D curve under different gate voltage recorded after various bending cycles. (c) On/off current measured in FeFET as a function of the number of bending cycles. These measurements were performed at $V_D = 0.1$ V with a bending radius of 7 mm. (d) Performance of the endurance property for the ITO FeFETs with different bend cycles.

Comment 4: The SS slope significantly below 60mV/dec does not appear to be consistently apparent. Can the authors support the evaluation with further measurement?

Response 4: Thank you for your insightful comment. We appreciate the importance of a low subthreshold swing (SS) as a crucial indicator of our transistor's efficiency and transfer characteristic steepness.

In our manuscript, we report an SS value significantly below the 60mV/dec limit,

which is indicative of the superior gate control and the reduced interface states in our device. To further substantiate these findings, we've conducted extensive measurements on an expanded set of 30 devices. The statistical distribution of these SS slopes, presented in Figure R21, strengthens the reliability of our results.

Notably, 80% of the devices tested—24 out of 30—exhibited SS values below 60mV/dec. The remaining six devices also demonstrated low subthreshold swings, with the highest only at 90mV/dec, validating the reproducibility and dependability of our reported SS values.

Figure R21 Statistical distribution of SS slopes for 30 different devices.

The device architecture plays a pivotal role in achieving these consistent SS values. The ultra-thin HZO ferroelectric layer, combined with the ITO channel, ensures excellent gate control crucial for a steep SS. This is further evidenced by the P-V characteristics in Figures R22a and R22a, which suggest a high-quality interface that suppresses trap-assisted tunneling and charge trapping effects.

Figure R22 P-V loops of representative capacitors with (a) W/11.4 nm HZO /Ni and (b) W/11.4 nm HZO/3.4 nm ITO/Ni. The similar P-V characteristics suggest that the interface of FE and channel is excellent.

We have included this additional data in the revised manuscript and supplementary information and highlighted it, reinforcing the robustness of our fabrication process and measurement protocol. We hope this augmented data set will adequately address your queries regarding the SS values.

Corresponding change in manuscript: Yes

Location of Change:

- On Page 8,

“The distribution of key performance indicators (KPIs) for 30 devices is presented in Supplementary Fig. 8, including critical performance information such as the on/off ratio, S.S., MW, and I_{ON} . The overall performance of the devices is favorable, which can be attributed to mature device fabrication processes and stable experimental conditions.”

- On page S7 of Supplementary Information,

Supplementary Fig.8 The distribution of key performance indicators (KPIs) for 30 devices. Distribution of (a) on/off ratio and (b) memory window for the devices. Statistics of (c) subthreshold swing (S.S.) and (d) gate leakage current (I_G). (e) Statistics of on-state current (I_{ON}).

Thanks a lot for your suggestions.

Expect for your appreciate reconsideration!

We hope the above improvements can meet the reviewer's comments, thank you very much for your careful reading.

REVIEWERS' COMMENTS

Reviewer #1 (Remarks to the Author):

The authors have taken considerations of all the concerns raised and improved the manuscript based on the suggestions. Therefore I recommend publication of the revised manuscript.

Reviewer #2 (Remarks to the Author):

The authors provided comprehensive and valuable answers to the reviewers' questions and took them into account accordingly in the manuscript. Many thanks for this support.

From my side no further comments and questions.

Reviewer #3 (Remarks to the Author):

Thank you for your prompt and comprehensive responses to my questions and comments regarding this paper. I appreciate the effort you've put into addressing each point raised in my review.

Overall, I find your responses satisfactory and am confident that the updated version of the paper will significantly contribute to the advancement of knowledge in this field.

Thank you once again for your responsiveness and dedication to improving your manuscript based on reviewer feedback.

Response Letter

(Black font: reviewers' original comments; Blue font: Our response; Red font: revision in the manuscript.)

Reviewer #1 (Remarks to the Author):

The authors have taken considerations of all the concerns raised and improved the manuscript based on the suggestions. Therefore I recommend publication of the revised manuscript.

Reviewer #2 (Remarks to the Author):

The authors provided comprehensive and valuable answers to the reviewers' questions and took them into account accordingly in the manuscript. Many thanks for this support. From my side no further comments and questions.

Reviewer #3 (Remarks to the Author):

Thank you for your prompt and comprehensive responses to my questions and comments regarding this paper. I appreciate the effort you've put into addressing each point raised in my review.

Overall, I find your responses satisfactory and am confident that the updated version of the paper will significantly contribute to the advancement of knowledge in this field.

Thank you once again for your responsiveness and dedication to improving your manuscript based on reviewer feedback.

Response:

We appreciate the thoughtful comments and questions/suggestions regarding our manuscript, " High-performance ferroelectric field-effect transistors with ultra-thin indium tin oxide channels for flexible and transparent electronics" (ID: NCOMMS-23-

56290A). We hope that the above improvements can convince the referees that our paper is suitable to be published in **Nature Communications**.